# MOTHERNET: FAST TRAINING AND INFERENCE VIA HYPER-NETWORK TRANSFORMERS

**Andreas C. Müller, Carlo Curino & Raghu Ramakrishnan**
Gray Systems Lab
Microsoft
{amueller,ccurino,raghu}@microsoft.com

## ABSTRACT

Foundation models are transforming machine learning across many modalities, with in-context learning replacing classical model training. Recent work on tabular data hints at a similar opportunity to build foundation models for classification for numerical data. However, existing meta-learning approaches can not compete with tree-based methods in terms of inference time. In this paper, we propose MotherNet, a hypernetwork architecture trained on synthetic classification tasks that, once prompted with a never-seen-before training set generates the weights of a trained "child" neural-network by in-context learning using a single forward pass. In contrast to most existing hypernetworks that are usually trained for relatively constrained multi-task settings, MotherNet can create models for multiclass classification on arbitrary tabular datasets without any dataset specific gradient descent. The child network generated by MotherNet outperforms neural networks trained using gradient descent on small datasets, and is comparable to predictions by TabPFN and standard ML methods like Gradient Boosting. Unlike a direct application of TabPFN, MotherNet generated networks are highly efficient at inference time. We also demonstrate that HyperFast is unable to perform effective in-context learning on small datasets, and heavily relies on dataset specific fine-tuning and hyper-parameter tuning, while MotherNet requires no fine-tuning or per-dataset hyper-parameters.

## 1 INTRODUCTION

Foundation models, i.e., large transformer-based (Vaswani et al., 2017) models trained on massive corpora, are disrupting machine learning in many areas such as natural language and reasoning tasks. These domains shifted from small task-specific models to large generic models with task-specific instructions via prompting and in-context learning. However, this shift has not yet reached tabular data, the most common data type in real-world machine learning applications (Chui et al., 2018), which is still dominated by traditional machine learning methods and neural networks with in-weight learning. This paper explores a new approach to applying transformer-based Foundational Models to tabular classification, demonstrating their potential to replace costly and slow AutoML with in-context learning. The existing TabPFN (Hollmann et al., 2022) approach is promising in terms of accuracy and training time, but falls short of state-of-the-art classical solutions in terms of training set scale, being restricted to 1000 to 3000 data points for training, and inference runtime, being approximately ten times slower to predict than a comparable tree-based method.

We introduce a new architecture, called MotherNet, which adapts the TabPFN architecture to a hypernetwork setup to produce model weights for a feed forward neural network. Our method performs competitively with baseline methods such as gradient boosting (Friedman, 2001; Chen & Guestrin, 2016) and outperforms learning a neural network with gradient descent, being much faster to train, providing higher accuracy and requiring no tuning of hyperparameters on small tabular datasets. Our approach combines the transformer architecture of TabPFN (Hollmann et al., 2022) with the idea of hypernetworks (Ha et al., 2017), to produce state-of-the-art classification models *in a single forward pass*. Unlike original work in hypernetworks (Ha et al., 2017), which used a small hyper network to generate a large "main" network, we are training a large, transformer-based hypernetwork to generate a compact classification network.

Compared to the approach of Hollmann et al. (2022), this allows for much shorter prediction time. However, just as `TabPFN`, `MotherNet` is restricted by the quadratic memory requirements of the transformer architecture, and does not scale well above approximately 5,000 data points. Compared to earlier work on hypernetworks, we train a single hypernetwork to address tabular classification on numeric data *in general*, i.e. in the style of a foundational model, instead of a task-specific or multi-task hypernetwork.

We demonstrate that it is possible to generate neural networks directly as the output of a transformer model, without the need to do any dataset-specific learning or gradient descent. Using a fixed model structure, we are able to produce neural networks that work well on small numeric tabular datasets from the OpenML CC-18 benchmark suite (Bischl et al., 2017), and show that our approach also provides a good trade-off of speed and accuracy on the TabZilla dataset collection McElfresh et al. (2024). Training and inference code and pre-trained model weights are made publicly available [1].

## 2    RELATED WORK

### 2.1    LLMS FOR TABULAR DATA

There have been several works investigating the use of small, heterogeneous tables for question answering, and extracting tables from data, using large language models (LLMs) and specifically fine-tuned transformers (Yin et al., 2020; Iida et al., 2021). These architectures are table-specific, but meant to answer natural language questions. Our work focuses on supervised classification. Separately approaches have been proposed to apply Large Language Models directly to supervised tabular classification tasks (Dinh et al., 2022; Hegselmann et al., 2023). These works generally require tokenization on the level of each input feature. This allows including world-knowledge about feature names and potentially feature values, but comes at an extreme computational cost, and strong limitations on the number of features and data points, as the size of the attention matrix is quadratic in both features and samples, and has a non-trivial constant factor for encoding floating point numbers into separated string values.

### 2.2    TABPFN

Recently Hollmann et al. (2022), building on the work of Müller et al. (2021), introduced a transformer architecture that is capable of performing supervised classification on tabular numeric data. This work is quite distinct from other transformer architectures on tabular data in that it is focused on numeric input and numeric output.

`TabPFN` uses a transformer where each input "token" is a row of the tabular dataset. The model is adapted to work with a variable number of features by zero-padding (and scaling) to 100 features. For the training data, linear projections of the input rows are summed with linear projections of integer classification labels. For the test data, the labels are omitted and class-probabilities are produced as output tokens. The model is trained to minimize cross-entropy on the test data points. Attention is masked so that all training points can attend to all other training points, while test points can only attend to training points. A variable number of classes is handled by training for up to ten classes, and when predicting for a dataset with $k \leq 10$ classes, using only the first $k$ outputs in the softmax layer. The authors design a prior over synthetically generated datasets, based on Structural Causal models and Bayesian Neural Networks. Using draws from this prior, they are able to train a model that generalizes to perform supervised classification on real-world tabular datasets. `TabPFN` showed strong predictive performance without any per-dataset tuning, and with extremely fast time to train and predict on small datasets ($\leq 3000$ data points) (McElfresh et al., 2024). Because of the quadratic nature of the self-attention matrix, training on larger datasets is impractical with the method proposed in Hollmann et al. (2022). Our work builds on the work of Hollmann et al. (2022), and adds the capability to create a dataset-specific model.

---

[1]`https://github.com/microsoft/ticl`

## 2.3 LEARNING TO LEARN AND META-LEARNING

There has been a long history of approaches to "learning to learn" and to build neural network that produce other neural networks (Schmidhuber, 1992; Ravi & Larochelle, 2016; Andrychowicz et al., 2016; Thrun & Pratt, 1998). Given the long history, we will only review some more recent and closely related approaches. Most approaches solve transfer-learning or task-adoption within a fairly narrow family of tasks, often one-dimensional regression or character recognition, while our work addresses classification on any small tabular dataset. Ha et al. (2017) introduce the term hypernetwork for networks that produce networks using task-specific embeddings. The hypernetwork and embedding are both learned via gradient descent on the same dataset, reducing the approach (in the non-recurrent case) to a standard neural network with a low-rank structure in its weights. Bertinetto et al. (2016) propose a gradient-free approach to produce student networks based on single-shot examples in OCR and object tracking. Their objective and formulation closely resembles ours; however, in this work, a single "exemplar" is a tabular training dataset, while in Bertinetto et al. (2016), it is a single handwritten digit, or a single object to track. A transformer based approach for a hypernetwork generating convolutional neural networks has been investigated in Zhmoginov et al. (2022). Conditional Neural Processes (Garnelo et al., 2018) also perform task adaption without gradient descent. In contrast to the original work of Garnelo et al. (2018), we are using a transformer architecture instead of a feed-forward neural network, and are able to address a much wider range of tasks. While later work on Neural Processes used transformers (Nguyen & Grover, 2022) in an architecture closely related to `TabPFN`, we are not aware of an implementation of Conditional Neural Processes using transformers, which would yield an architecture more similar to `MotherNet`. Most recently, Bonet et al. (2023) introduced `HyperFast`, a hypernetwork that, similar to `MotherNet`, is trained to perform classification on generic tabular datasets. `HyperFast` avoids the quadratic complexity of the transformer attention mechanism, and therefore scales to larger datasets. We compare to `HyperFast` with and without gradient descent and with hyper-parameter tuning in Section 4. and find that despite the large overlap of the `HyperFast` training set and our test-set, `HyperFast` without gradient descent performs poorly, and hyper-parameter tuning is necessary for good performance.

## 3 METHODOLOGY

Hollmann et al. (2022) introduced `TabPFN`, an adaption of the transformer architecture to solve tabular classification problems. As this work closely builds on `TabPFN`, we want to briefly review its architecture. `TabPFN` uses a transformer where each input "token" is a row of the tabular dataset. The model is adapted to work with a variable number of features by zero-padding (and scaling) to 100 features. For the training data, linear projections of the input rows are summed with linear projections of integer classification labels. For the test data, the labels are omitted and class-probabilities are produced as output tokens. The model is trained to minimize cross-entropy on the test data points. Attention is masked so that all training points can attend to all other training points, while test points can only attend to training points. A variable number of classes is handled by training for up to ten classes, and when predicting for a dataset with $k \leq 10$ classes, using only the first $k$ outputs in the softmax layer. `TabPFN` showed strong predictive performance without any per-dataset tuning, and with extremely fast time to train and predict on small datasets ($\leq 3000$ data points) (McElfresh et al., 2024). Because of the quadratic nature of the self-attention matrix, training on larger datasets is impractical with the method proposed in Hollmann et al. (2022).

**Limitations of `TabPFN`** Comparing speed and computational efficiency between `TabPFN` and traditional ML and AutoML methods is somewhat complicated, as they have very different characteristics. In particular, there is no dataset specific training phase after meta-training when applying `TabPFN`, only near-instantaneous in-context learning. Prediction, on the other hand, is significantly slower than prediction in standard ML models, as prediction requires computing attention between test and training data. On the other hand, gradient boosted trees (Friedman, 2001; Chen & Guestrin, 2016; Ke et al., 2017) have significant training cost, in particular when accounting for hyper-parameter tuning, but are extremely fast for prediction. Together with the memory requirements of a large transformer model, this makes `TabPFN` impractical for settings where fast, on-demand predictions are required. Next we present two approaches, an effective baseline distil-

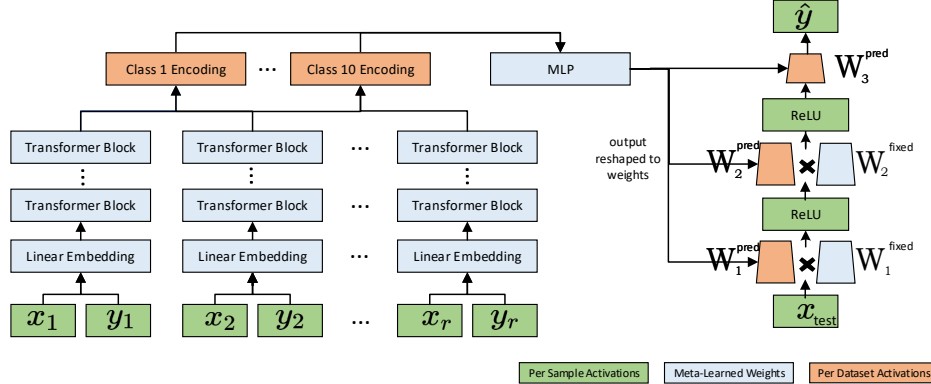

Figure 1: MotherNet architecture. Given training data $(x_1, y_1), \dots, (x_r, y_r)$, the transformer produces a vector $\phi$, which is reshaped to weight matrices of an MLP with low-rank weight structure. Green blocks are individual data points and their activations, orange layers are activations created during in-context learning or in the forward-pass during meta-learning, and light blue layers are learned during meta-training.

lation approach, and MotherNet, that address some of the runtime and scalability limitations of TabPFN.

## 3.1 MotherNet: Generating Model Weights

Motivated by the success of TabPFN, and inspired by previous work on hypernetworks, we propose MotherNet, a transformer architecture that is trained to produce machine learning models with trained weights in a single forward pass. This methodology combines the benefits of a Foundation Model that does not require dataset specific training or tuning with the high efficiency of a compact model at inference time. The resulting models are small feed-forward neural networks (an MLP with two hidden layers of size 512 in our experiments) that have competitive performance, created without the use of back-propagation or any loss minimization on the training set. The training process of the overall architecture can be described as:

$$\min_\theta \sum_i \mathcal{L}(\text{MLP}_\phi, D_i^p),$$

$$\text{where } \phi = \text{MotherNet}(D_i^t, \theta)) \tag{1}$$

Where $\theta$ are the parameters of the MotherNet transformer, $D_i^t$ and $D_i^p$ are training and prediction portion of a dataset $i$, $\text{MLP}_\phi$ is the feed-forward neural network with parameters given by $\phi$ and $\mathcal{L}(M, D)$ is the cross entropy loss of the model $M$ evaluated on datasets $D$. The model architecture is shown in Figure 1. Training is performed by back-propagation through the whole architecture (from the output of the child model, and through the transformer layers) where each training sample corresponds to one dataset. During this meta-training, the parameters $\theta$ are learned using synthetic datasets, using the prior from Hollmann et al. (2022) and then frozen. To apply the model to a new (real) dataset $\hat{D}$ consisting of a training portion $\hat{D}^t$ and a prediction portion $\hat{D}^p$, we evaluate MotherNet($\hat{D}^t, \theta$), which produces a vector of parameters $\hat{\phi}$. This vector is then used as the weight and bias vectors of a feed-forward neural network (properly reshaped), which can be used to make predictions on $\hat{D}^p$. We refer to this approach of applying MotherNet to create a child network as in-context learning. The absence of per-dataset gradient descent in our method not only provides an advantage in terms of runtime complexity, it also eliminates the need to apply and tune regularization, as the model was trained directly for *generalization*, similar to a Conditional Neural Process (Garnelo et al., 2018).

MotherNet maintains the structure of input encoding and twelve transformer layers of TabPFN on the training set, which produces embeddings of size $m$ (512 for the experiments) for each pair of

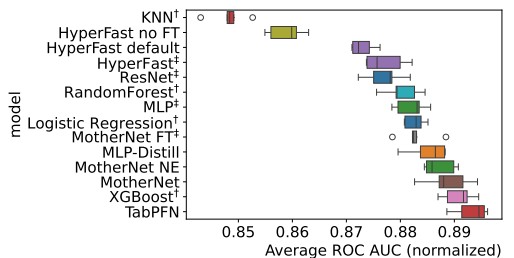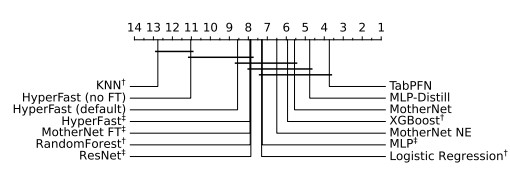

Figure 2: Comparison of `TabPFN`, `HyperFast`, `MLP-distill` and `MotherNet` with tuned baselines over the test datasets of Hollmann et al. (2022), listed in Table 8. † means 1h of HPO on CPU, ‡ means 1h of HPO on GPU. Left: Comparison of normalized mean ROC AUC. Right: Critical Differences Diagram (Demšar, 2006).

training data point and label. We use a one-hot encoding of classes, unlike Hollmann et al. (2022) who use a linear layer. We use the training labels to compute the average embedding per class, reducing all activations to a single dataset embedding $E$ of size $m_{\text{all}}$ ($512 \cdot 10$ in the experiments). This embedding $E$ is decoded into the vector $\phi$ using a one-hidden-layer feed-forward neural network. The vector $\phi$ of activations based on the transformer is then reshaped into weights and biases for the "child" feed-forward network. We evaluated different variants of the architecture hyper-parameters on the validation set of Hollmann et al. (2022). Initial experiments showed this approach to be very successful, but lead to very large transformer models as a function of the size of the network that was to be produced. To reduce model size, we decomposed the weights into two components, $\mathbf{W}_i^{\text{p}}$ that is produced as a prediction by the transformer, and $\mathbf{W}_i^{\text{f}}$ that is learned during the meta-training phase and fixed during in-context learning, similar to Ha et al. (2017).

The low-rank structure drastically reduces the number of entries in $\phi$ for a given size of neural network produced. All our experiments use the low-rank version, which yielded slightly better AUC on the validation set, at a much smaller model size. As output architecture, we use an MLP with two hidden layers with 512 hidden units each, and with weight-matrices of rank 32. More specifically, for rank $r = 32$, hidden dimension $h = 512$, $d = 100$ features and $N = 10$ classes, with $\mathbf{W}_1^{\text{p}}, \mathbf{W}_2^{\text{p}} \in \mathbb{R}^{h \times r}$, and $\mathbf{W}_3^{\text{p}} \in \mathbb{R}^{N \times r}$ produced by the transformer, and $\mathbf{W}_1^{\text{f}} \in \mathbb{R}^{r \times d}$ and $\mathbf{W}_2^{\text{f}} \in \mathbb{R}^{r \times h}$ learned during meta-learning, predictions are made as

$$\mathbf{h_1} = \text{relu}(\mathbf{W}_1^{\text{p}}\mathbf{W}_1^{\text{f}}\mathbf{x}) \qquad \mathbf{h_2} = \text{relu}(\mathbf{W}_2^{\text{p}}\mathbf{W}_2^{\text{f}}\mathbf{h_1}) \qquad p(\hat{y}) = \text{softmax}(\mathbf{W}_3^{\text{p}}\mathbf{h_2})$$

The best-performing version of our architecture has 89M parameters, with 63M of these in the decoder attached. Somewhat surprisingly, we found that a decoding MLP with a hidden layer of size of 4096 works well. This means the whole training dataset is first compressed to a vector of length 4096, and then expanded into a vector of size $25,738$ to encode the low-rank components of the weights for the network that is produced. We train `MotherNet` on a single A100 GPU with 80GB of GPU memory, which takes approximately four weeks. We are using increasing batch sizes of 8, 16 and 32 and a learning rate of 0.00003, with cosine annealing (Loshchilov & Hutter, 2016).

## 3.2 In-Context Learning with `MotherNet`

For in-context learning on a new a dataset $D^t$ with $r$ features and $c$ classes, we perform a forward pass in the transformer to obtain $\phi = \texttt{MotherNet}(\mathbf{D}_i^{\text{t}}, \theta)$. We discard all but the first $r$ rows of the input layer matrix $\mathbf{W}_1^{\text{f}}$, and keep only the first $c$ columns of the output matrix $\mathbf{W}_3^{\text{p}}$, resulting in a network with an input layer of size $r$, hidden layers of size 512 and an output layer of size $c$. Because of the quadratic complexity of full attention, the size of training dataset that is feasible to process is limited by available memory. We were able to process up to 30,000 data points on an A100 GPU with 80GB of memory, and 100,000 samples on CPU. However, we did not evaluate accuracy on datasets of this size. We found that the ensembling strategy of Hollmann et al. (2022) improves predictive performance. We apply a similar strategy, using different circular permutations of features and classes, optionally using one-hot-encoding for categorical variables and optionally using quantile encoding for continuous features. For this larger ensembling we sample 8 models

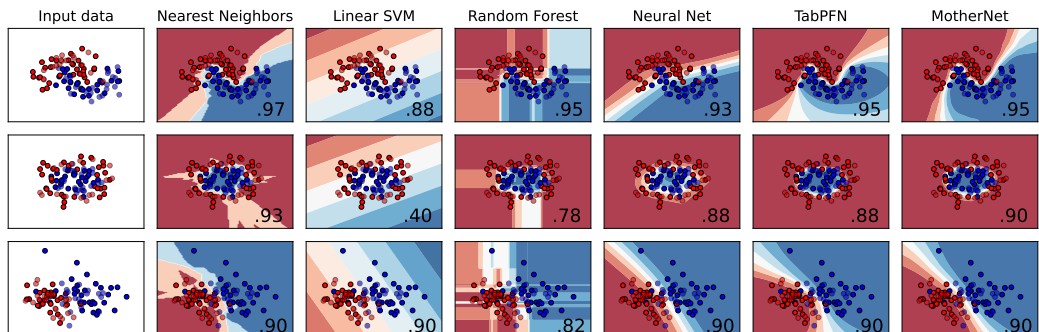

Figure 3: Comparing decision boundaries on synthetic toy datasets, adapted from the `scikit-learn` (Pedregosa et al., 2011) documentation. `MotherNet` decision boundaries closely resemble `TabPFN` and to a lesser degree traditional Neural Network boundaries.

from this space for all our experiments; for `TabPFN` we follow the default setting of 3 from Hollmann et al. (2022), which seems sufficient for that model. Sampling a larger number for either improves performance but slows down both training and predictions with diminishing returns. The predictions that are produced, both by individual networks, and the ensemble, are quite smooth and similar to traditionally learned neural networks or `TabPFN`, see Figure 3.

### 3.3 DISTILLATION BASELINE

To get a better understanding of the limitations and trade-offs inherent in the `MotherNet` architecture, we also investigate a baseline approach for creating a small neural network based on `TabPFN` via distillation. Distillation is a natural approach to reducing prediction time, while making use of the excellent performance of `TabPFN`. Distillation is a slower and less direct way to extract a dataset specific model from the `TabPFN` approach than using `MotherNet`. However, it allows us to disentangle the contribution of model capacity, the ability of the hypernetwork to create appropriate weights, and the predictive bias of the `TabPFN` training procedure. We apply the predictions of `TabPFN` as a teacher model for a small feed-forward neural network, that is trained specifically for a dataset, analogous to the methodology of Hinton et al. (2015). Note that we are not attempting to distill `TabPFN`, but instead create dataset-specific distillations for each dataset we want to predict on. Furthermore, while `TabPFN` is acting as a teacher model, it never saw the dataset for which it is a teacher during training time. While this is a natural way to distill the model for dataset-specific prediction, we are not aware of this being investigated before. Tuning the distilled model architecture on the validation set of Hollmann et al. (2022) found that a relatively small model suffices for good performance, leading to a reduction in size of the model of nearly 3 orders of magnitude: We use an MLP with two hidden layers of size 128, which results in a maximum of $\sim 30k$ parameters (with 100 input features and 10 targets, i.e., the limit in the datasets we consider as most datasets are smaller), while the original `TabPFN` has $\sim 26M$ parameters.

## 4 EXPERIMENTAL EVALUATION

We evaluate `MotherNet` on two tabular benchmarks, the small datasets in OpenML CC18, as used by Hollmann et al. (2022) and a version of the TabZilla benchmark (McElfresh et al., 2024). As previous work has shown, selection of the benchmark can have a large effect on the ranking of algorithms; therefore, any ranking can only be relative to a given benchmark. We use two benchmarks from the literature to show that `MotherNet` has competitive predictive performance, while maintaining a unique combination of no hyper-parameter tuning, extremely efficient (in context) learning, and efficient prediction. All our experiments were done on a A100 GPU with 80GB of RAM on cloud infrastructure.

| model | rank | normalized AUC | AUC | fit time (s) | predict time (s) | fit + predict |
|---|---|---|---|---|---|---|
| TabPFN | 3.733 | 0.850±0.026 | 0.893±0.003 | 0.008 | 0.337 | 0.344 |
| MLP-Distill | 4.767 | 0.802±0.027 | 0.885±0.004 | 4.382 | 0.002 | 4.384 |
| MotherNet | 5.567 | 0.780±0.037 | 0.889±0.004 | 0.136 | 0.006 | 0.143 |
| MotherNet (CPU) | | | | 7.904 | 0.107 | 8.010 |
| XGBoost[†] | 5.933 | 0.804±0.027 | 0.891±0.003 | 4.912 | 0.024 | 4.936 |
| MotherNet NE | 6.500 | 0.716±0.026 | 0.887±0.003 | 0.040 | 0.001 | 0.040 |
| MLP[‡] | 7.267 | 0.744±0.025 | 0.882±0.003 | 2.257 | 0.001 | 2.258 |
| Logistic Regression[†] | 7.300 | 0.749±0.024 | 0.883±0.002 | 0.209 | 0.000 | 0.209 |
| ResNet[‡] | 7.867 | 0.710±0.025 | 0.877±0.004 | 1.668 | 0.001 | 1.669 |
| RandomForest[†] | 7.900 | 0.732±0.020 | 0.880±0.003 | 0.200 | 0.032 | 0.232 |
| MotherNet FT[‡] | 7.900 | 0.708±0.024 | 0.883±0.004 | 0.839 | 0.002 | 0.841 |
| HyperFast[‡] | 7.900 | 0.729±0.028 | 0.877±0.004 | 15.931 | 0.083 | 16.014 |
| HyperFast default | 8.567 | 0.681±0.020 | 0.873±0.002 | 25.960 | 0.046 | 26.006 |
| HyperFast no FT | 11.033 | 0.530±0.030 | 0.859±0.003 | 3.792 | 0.045 | 3.837 |
| KNN[†] | 12.767 | 0.453±0.021 | 0.848±0.003 | 0.000 | 0.007 | 0.008 |

Table 1: Summary results on small CC-18 datasets. Average rank is based on normalized ROC AUC. ± denotes std over the five paired splits of each dataset. † means 1h of HPO on CPU, ‡ means 1h of HPO on GPU. Note that `HyperFast` and `ResNet` use $1h$ of GPU tuning time per dataset, while `MotherNet` requires a single fit that take 0.14 seconds on average.

## 4.1 OPENML CC18 (SMALL)

We follow the experimental evaluation of Hollmann et al. (2022), and focus our evaluation on the 30 datasets within the CC-18 with less than 2000 samples, listed in Table 8 in the appendix, and compare models using one-vs-rest ROC AUC. When aggregating across datasets, we normalize scores with the minimum and maximum performance achieved on the dataset by any algorithm. As in Hollmann et al. (2022), we split each dataset 50/50 into training (or in-context learning) and test set, and repeat this split five times. We refer to that work for an in-depth comparison of `TabPFN` with current AutoML methods. We compare predictive performance and runtime of the following models: `TabPFN` as provided by the authors of Hollmann et al. (2022), `HyperFast` (Bonet et al., 2023), a recent hyper-network architecture for tabular classification, `MLP-distill`, `MotherNet`, a vanilla MLP, a ResNet using the architecture of Gorishniy et al. (2021) and baselines consisting of Histogram Gradient Boosting (Chen & Guestrin, 2016; Friedman, 2001), $k$-Nearest Neighbors, Logistic Regression and Random Forests Breiman (2001). In contrast to Hollmann et al. (2022), we include datasets which contain categorical features or missing values; we fill missing values with zeros as in Hollmann et al. (2022). We perform no dataset specific tuning for the transformer architectures, while baseline approaches use 60 minutes hour of randomized hyper-parameter tuning.

Quantitative results are shown in Figure 2 and Table 4, where errors are given over the five paired splits of the data. We can see that `TabPFN` outperforms all other methods, though not statistically significantly so, even at 60 minutes of tuning time for reference methods. While the distilled version `MLP-distill` does not achieve the same level of performance, it outperforms all the baseline models even without dataset specific tuning. It seems the probabilistic predictions produced by `TabPFN` provide enough regularization for good generalization. Our `MotherNet` outperforms all the baseline approaches, and outperforms `MLP-distill` in terms of normalized ROC AUC, but is outperformed by `MLP-distill` in terms of rank. Results using the validation set of Hollmann et al. (2022) can be found in the appendix in Figure 4. Comparing with the recent `HyperFast` (Bonet et al., 2023) it should be noted that of the 30 datasets we use for evaluation, 18 are in the `HyperFast` training set, giving it a direct advantage. We find that without per-dataset finetuning `HyperFast` (`HyperFast` no FT) does not provide comparable performance to any of the baseline approaches except KNN. Using per-dataset fine-tuning and the default hyper-parameters `HyperFast` (`HyperFast` default) is also outperformed by the baselines. Using 60 minutes of randomized hyper-parameter tuning on GPU, `HyperFast` is competitive with `XGBoost`, but outper-

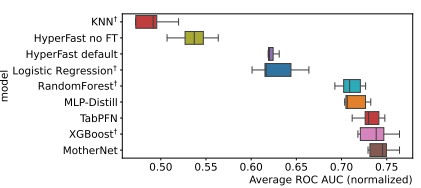 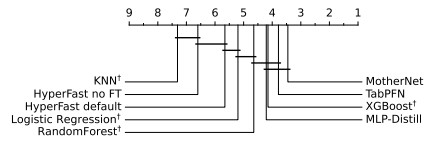

Figure 4: Comparison of `TabPFN`, `HyperFast`, `MLP-distill` and `MotherNet` with tuned baselines over the validation datasets of Hollmann et al. (2022), listed in Table 9. † means 1h of HPO on CPU, ‡ means 1h of HPO on GPU. Left: Comparison of normalized mean ROC AUC. Right: Critical Differences Diagram (Demšar, 2006). Compare with Figure 2 for the test set. We did not include `HyperFast`, `MLP` and `ResNet` because of the extreme computational cost of tuning these models for each dataset.

formed by `MotherNet`, `MLP-distill` and `TabPFN`, despite a large fraction of the test datasets being included in the `HyperFast` training. Also, note that 1h of per-dataset hyper-parameter tuning on GPU corresponds to approximately 25,000x more compute than `MotherNet`, which requires no fine-tuning or hyper-parameter tuning and trains within 0.14s on average on the test datasets. Interestingly, the MLP which is trained using standard gradient descent is performing much worse than `MLP-distill` and `MotherNet` despite extensive hyper-parameter tuning. Clearly `MotherNet` provides an efficient and easy-to-use alternative to training with gradient descent on these datasets, and for the small dataset regime that we investigate, hyper-parameter tuning can be difficult. It's noteworthy that on the OpenML CC-18 collection, Logistic Regression performs surprisingly well. This is likely a consequence of the dataset selection. Compare Figure 4 for the validation set, which has a more typical ranking of algorithms.

To determine whether fine-tuning is beneficial for models produced by `MotherNet`, we perform an experiment in which we apply gradient descent to the child model on the training dataset that was used for in-context learning. Since fine-tuning for all ensemble members would be costly, we compare `MotherNet` without ensembling (`MotherNet` NE) with the fine-tuned `MotherNet` (`MotherNet` FT). We tune learning rate, weight decay, use of dropout, number of epochs, and whether or not to use one-hot-encoding (for both in-context learning and gradient descent). We were unable to improve results using careful fine-tuning using 1h of HPO on GPU. Search produced either results that were equivalent to the MLP model, ignoring the `MotherNet` initialization, or left the initialization unchanged. This is not entirely surprising: in most cases, `MotherNet` improves over the MLP model, and we hypothesize that this improvement stems from a learned regularization performed by the hyper-network; in particular for small datasets, overfitting is a major issue for neural architectures, and tuning hyper-parmeters is difficult since validation set results are noisy. Applying gradient descent in this setting nullifies the benefits of the learned regularization. This might no longer hold when using larger datasets, which we plan to investigate in future work.

Regarding algorithm speed, if we only consider prediction time, `TabPFN` on GPU is about ten times slower than `XGBoost`, while `MotherNet` on GPU is about five times faster than `XGBoost`, or 50 times faster than `TabPFN`. This is the main advantage we look to gain from using `MotherNet` over `TabPFN`. However, `MLP-distill` is even faster, at about 3x the speed of `MotherNet`, likely due to the ensembling described in Section 3.2. However, if we look at the speed of training (assuming optimum hyper-parameters are known) together with prediction, a measurement particularly critical for model development, we see that `MotherNet` on GPU provides a 33x speedup over `XGBoost`, while `MLP-distill` is over 30x slower than `MotherNet`. See Figure 5 (left) for a visualization of the speed-AUC trade-off.

In most real-world settings, hyper-parameters are unknown, results in Figure 2 and Table 4 shows that even with 1h of hyper-parameter tuning, the baseline models underperform the transformer models. Taking this tuning time into consideration, using `MotherNet` on the GPU results in more than 25,000x speedup for model-development, while `MotherNet` on the CPU still provides 450x speedup. We want to point out that these speedup figures might depend strongly on the tuning time estimated for other algorithms. This points at a practical difficulty of tuning parameters: in practice it

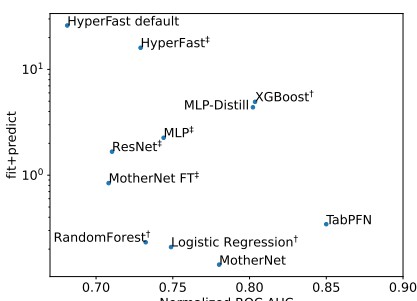 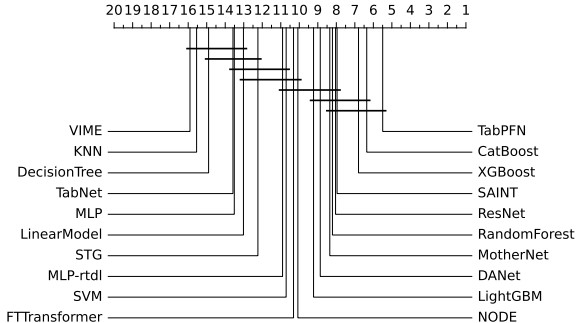

Figure 5: Left: Visualizing runtime vs normalize AUC trade-off, based on numbers in Table 4, not considering tuning times for the competing algorithms. Bottom right means fast and accurate algorithms. Note that the y-axis is in log-scale. Right: Critical Difference Diagram on the TabZilla benchmark using ROC AUC, corresponding to results in Table 2.

is often unclear how much time should be allocated for hyper-parameter tuning. Using `MotherNet` removes this trade-off by providing competitive accuracy near-instantaneously without any tuning. A more detailed comparison between `MLP-distill`, `MotherNet` and `TabPFN` can be found in Table 3 and Table 4. We see that on some datasets, `MLP-distill` performs much worse than `MotherNet`, likely because of the sensitivity of gradient descent to hyper-parameters. Detailed results can be found in Table 3 in Appendix A.

## 4.2 TABZILLA

We use the TabZilla (McElfresh et al., 2024) benchmark to compare to a wide varity of deep learning and tree-based approached. Table 4.2 reproduces the results of McElfresh et al. (2024), with our results for `MotherNet` added. For this evaluation, we follow McElfresh et al. (2024) in their setup for `TabPFN`, and subsample 3000 data points for `MotherNet`, as the full datasets are too large for the transformer architectures. This means that both `MotherNet` and `TabPFN` have a severe disadvantage, as they only see a fraction of the data provided to other algorithms. Despite this disadvantage, `TabPFN` still outranks other algorithms. `MotherNet` is outranked by some of the tree-based learners, as well as `SAINT` and `ResNet` in rank, though `MotherNet`, `SAINT` and `ResNet` have equivalent mean normalized AUC. The critical difference diagram using ROC-AUC can be found in Figure 5 (right), and more results can be found in Table 7 and Table 6 in the Appendix. The CD diagram shows no significant differences between the top seven algorithms, despite the fact that other algorithms were given up to 10h of compute and up to 30 hyper-parameter sets, while `MotherNet` requires no hyper-parameter tuning and finishes within seconds on all datasets. The combined training and prediction time of `MotherNet` is competitive with those of the tree-based models (though comparing `MotherNet` on GPU with tree-based models on CPU) and orders of magnitude faster than other deep learning approaches.

## 5 LIMITATIONS AND FUTURE WORK

One of many open questions is to understand how the models produced by `MotherNet` differ from those produced by gradient descent. The nature of the optimization is fundamentally different, and in essence, `MotherNet` learns to regularize according to the datasets presented during meta-training, instead of using a hard-coded regularization strategy such as weight decay or early stopping. We were able to achieve good performance with a single neural network architecture across all datasets, both for `MotherNet` and `MLP-distill` (though with slightly different architectures for the two), which seems hard to achieve with standard gradient descent. The relative performance of `MLP-distill`, `TabPFN` and `MotherNet` is somewhat muddled, and inconsistent between the test set and validation set, see figures Figure 2 and Figure 4, and between mean AUC and rank. We found that using one-hot-encoding is critical for the prediction network produced by `MotherNet` to perform well, an issue that is not present in the `TabPFN` architecture, and neces-

| model | rank | normalized AUC | fit time (s) | predict time (s) | fit + predict (s) |
|---|---|---|---|---|---|
| TabPFN | 5.10 | 0.91±0.17 | 0.25 | 16.13 | 16.38 |
| CatBoost | 5.99 | 0.92±0.18 | 20.75 | 0.05 | 20.80 |
| XGBoost | 6.55 | 0.90±0.19 | 0.85 | 0.04 | 0.89 |
| ResNet | 7.65 | 0.84±0.19 | 15.95 | 1.61 | 17.56 |
| SAINT | 7.66 | 0.84±0.19 | 171.13 | 2.56 | 173.69 |
| RandomForest | 7.93 | 0.87±0.19 | 0.41 | 0.56 | 0.97 |
| MotherNet | 8.30 | 0.84±0.18 | 0.34* | 0.11* | 0.45* |
| DANet | 8.65 | 0.83±0.19 | 64.50 | 1.32 | 65.82 |
| LightGBM | 9.20 | 0.83±0.22 | 0.89 | 0.04 | 0.93 |
| NODE | 9.77 | 0.81±0.20 | 161.05 | 1.81 | 162.86 |
| FTTransformer | 10.00 | 0.79±0.20 | 27.88 | 1.85 | 29.73 |
| SVM | 10.52 | 0.75±0.22 | 61.21 | 0.24 | 61.45 |
| MLP-rtdl | 10.61 | 0.73±0.21 | 15.18 | 1.57 | 16.75 |
| STG | 12.01 | 0.66±0.24 | 18.66 | 0.03 | 18.69 |
| Logistic Regression | 12.79 | 0.62±0.23 | 0.04 | 0.01 | 0.05 |
| MLP | 13.30 | 0.65±0.23 | 18.32 | 1.48 | 19.80 |
| TabNet | 13.50 | 0.63±0.32 | 35.19 | 0.61 | 35.80 |
| DecisionTree | 14.68 | 0.53±0.30 | 0.02 | 0.01 | 0.03 |
| KNN | 15.40 | 0.52±0.25 | 0.01 | 0.42 | 0.43 |
| VIME | 15.84 | 0.49±0.30 | 17.90 | 1.45 | 19.35 |

Table 2: Ranking of algorithms on TabZilla dataset collection, using normalized ROC AUC. As datasets have widely varying sizes, times are per 1000 data points. *Indicated experiments run for this paper, which are run on a A100 GPU as opposed to a V100 GPU as used by McElfresh et al. (2024) to produce the other timing results.

sitates additional bagging for prediction. In future work, we hope to address this issue directly in the architecture. There are also certain failure cases that `TabPFN` and `MotherNet` share, which are discussed in Appendix B. Another area of exploration is scaling the `MotherNet` method to larger training datasets. As mentioned above, the transformer architecture does not scale well in number of datapoints, and we focus our evaluation on training sets with 3000 samples or fewer. There is substantial literature on improving the complexity of attention mechanisms (see (Tay et al., 2022) for an overview), as well as more recent work into attention-free architectures (Fu et al., 2023; Poli et al., 2023). Both are promising candidates for scaling `MotherNet` to larger dataset sizes.

## 6 CONCLUSION

We demonstrated that it is possible to achieve competitive results on small numeric tabular datasets by producing neural networks using in-context learning via a single forward pass in our `MotherNet` architecture, without using dataset specific gradient descent or hyper-parameter tuning. By employing a pure meta-learning approach, similar to Conditional Neural Processes, we remove the need for explicit regularization, and therefore eliminate the hyper-parameters usually associated with learning neural networks. Compared to `TabPFN`, prediction speed on the test set is much faster, and training and prediction speed are both comparable to highly optimized tree-based models. We also find that distilling `TabPFN`, into a small neural network is highly effective and doesn't require dataset-specific hyper-parameter tuning — as opposed to training a similar neural network from scratch. Our work outperforms the other recent hyper-network architecture `HyperFast` on small datasets, both in accuracy and runtime, while not requiring dataset-specific gradient descent or hyper-parameter tuning, and despite our test set having large overlap with the `HyperFast` training set. The fact that competitive models can be generated with a simple forward pass is quite surprising, and opens up a new direction for producing high-performance models with fast inference using deep learning techniques.

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

| dataset | Hyper Fast‡ | KNN† | Log Reg† | MLP‡ | MN | MN NE | MN FT‡ | RF† | ResNet‡ | TabPFN | XGB† |
|---|---|---|---|---|---|---|---|---|---|---|---|
| MiceProtein | 1.00 | 1.00 | 1.00 | 1.00 | 1.00 | 1.00 | 1.00 | 1.00 | 1.00 | 1.00 | 1.00 |
| analcatdata authorship | 1.00 | 1.00 | 1.00 | 1.00 | 1.00 | 1.00 | 1.00 | 1.00 | 1.00 | 1.00 | 1.00 |
| analcatdata_dmft | 0.56 | 0.55 | 0.57 | 0.57 | 0.57 | 0.56 | 0.55 | 0.59 | 0.54 | 0.58 | 0.57 |
| balance-scale | 0.99 | 0.89 | 0.96 | 0.99 | 0.99 | 0.99 | 0.99 | 0.84 | 0.97 | 1.00 | 0.99 |
| banknote-authentication | 1.00 | 1.00 | 1.00 | 1.00 | 1.00 | 1.00 | 1.00 | 1.00 | 1.00 | 1.00 | 1.00 |
| blood-transfusion-service-center | 0.73 | 0.71 | 0.75 | 0.73 | 0.76 | 0.76 | 0.76 | 0.72 | 0.74 | 0.76 | 0.74 |
| breast-w | 0.99 | 0.98 | 0.99 | 0.99 | 0.99 | 0.99 | 0.99 | 0.99 | 0.99 | 0.99 | 0.99 |
| car | 0.99 | 0.92 | 0.98 | 1.00 | 0.97 | 0.97 | 0.98 | 0.99 | 1.00 | 1.00 | 1.00 |
| climate-model simulation crashes | 0.90 | 0.85 | 0.93 | 0.91 | 0.95 | 0.94 | 0.93 | 0.87 | 0.92 | 0.94 | 0.93 |
| cmc | 0.69 | 0.63 | 0.68 | 0.67 | 0.73 | 0.72 | 0.71 | 0.73 | 0.68 | 0.73 | 0.73 |
| credit-approval | 0.93 | 0.91 | 0.92 | 0.92 | 0.93 | 0.93 | 0.93 | 0.94 | 0.91 | 0.93 | 0.94 |
| credit-g | 0.76 | 0.73 | 0.77 | 0.76 | 0.79 | 0.79 | 0.79 | 0.79 | 0.76 | 0.79 | 0.79 |
| cylinder-bands | 0.81 | 0.78 | 0.82 | 0.84 | 0.83 | 0.83 | 0.80 | 0.87 | 0.83 | 0.83 | 0.89 |
| diabetes | 0.81 | 0.81 | 0.84 | 0.84 | 0.84 | 0.84 | 0.83 | 0.83 | 0.83 | 0.84 | 0.84 |
| dresses-sales | 0.55 | 0.56 | 0.57 | 0.57 | 0.59 | 0.61 | 0.58 | 0.56 | 0.56 | 0.54 | 0.59 |
| eucalyptus | 0.87 | 0.80 | 0.90 | 0.89 | 0.93 | 0.92 | 0.90 | 0.90 | 0.89 | 0.92 | 0.90 |
| ilpd | 0.70 | 0.65 | 0.74 | 0.73 | 0.73 | 0.73 | 0.70 | 0.71 | 0.70 | 0.74 | 0.71 |
| kc2 | 0.80 | 0.78 | 0.83 | 0.81 | 0.83 | 0.83 | 0.83 | 0.83 | 0.81 | 0.83 | 0.82 |
| mfeat-fourier | 0.98 | 0.97 | 0.98 | 0.97 | 0.97 | 0.97 | 0.97 | 0.98 | 0.98 | 0.98 | 0.98 |
| mfeat-karhunen | 1.00 | 0.99 | 1.00 | 1.00 | 0.99 | 0.97 | 0.99 | 1.00 | 1.00 | 1.00 | 1.00 |
| mfeat-morphological | 0.97 | 0.95 | 0.97 | 0.97 | 0.96 | 0.96 | 0.96 | 0.96 | 0.97 | 0.97 | 0.96 |
| mfeat-zernike | 0.98 | 0.98 | 0.98 | 0.98 | 0.98 | 0.97 | 0.97 | 0.97 | 0.98 | 0.98 | 0.97 |
| pc1 | 0.82 | 0.78 | 0.83 | 0.80 | 0.83 | 0.85 | 0.84 | 0.84 | 0.79 | 0.87 | 0.84 |
| pc3 | 0.82 | 0.75 | 0.79 | 0.80 | 0.81 | 0.81 | 0.80 | 0.82 | 0.79 | 0.84 | 0.82 |
| pc4 | 0.90 | 0.82 | 0.89 | 0.90 | 0.93 | 0.92 | 0.91 | 0.92 | 0.88 | 0.94 | 0.93 |
| qsar-biodeg | 0.92 | 0.89 | 0.92 | 0.92 | 0.92 | 0.92 | 0.92 | 0.92 | 0.92 | 0.93 | 0.92 |
| steel-plates-fault | 0.95 | 0.92 | 0.94 | 0.95 | 0.94 | 0.91 | 0.93 | 0.96 | 0.95 | 0.96 | 0.96 |
| tic-tac-toe | 0.92 | 0.99 | 1.00 | 1.00 | 0.99 | 1.00 | 1.00 | 0.98 | 1.00 | 0.96 | 1.00 |
| vehicle | 0.95 | 0.88 | 0.95 | 0.96 | 0.95 | 0.94 | 0.94 | 0.92 | 0.95 | 0.96 | 0.93 |
| wdbc | 0.99 | 0.99 | 0.99 | 0.99 | 1.00 | 1.00 | 1.00 | 0.99 | 0.99 | 1.00 | 0.99 |

Table 3: Per dataset results on the test set for small CC-18 datasets, averaged over 5 splits. † means 1h of HPO on CPU, ‡ means 1h of HPO on GPU. MN is `MotherNet`, MN no bag means a single network without bagging, and MN GD means a single network without bagging, with additional per-dataset gradient descent. It's unclear how to combine bagging and gradient descent here, which is why we consider comparison against the unbagged model.

# Appendices

APPENDIX A   PER-DATASET COMPARISON ON THE TEST SET

We show a per-dataset comparison of average ROC AUC of `TabPFN`, `MotherNet`, `MLP-distill` and `XGBoost` in Table 3. In contrast to the validation set, there seems to be no clear winner between `MLP-distill` and `MotherNet`. Overall, it seems hard to determine overall trends, but it's likely that dataset characteristics play a role, as we can observe similar relative performance in `eucalyptus`, `dresses-sales` and `cylinder-bands`, while the `mfeat` datasets and `MiceProtein` show a very different profile. We plan to revisit this comparison after addressing the issues discussed in Section B.

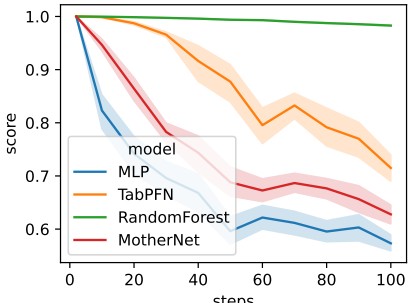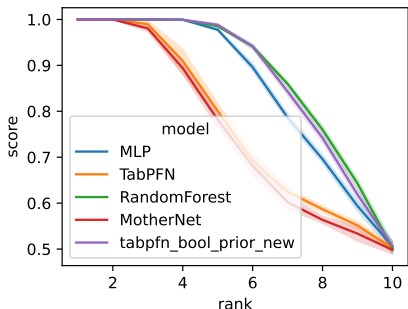

Figure 6: Mean test-set accuracy on synthetic binary classification datasets comparing `TabPFN` and `MotherNet` to untuned variants of `scikit-learn` classifiers. Left: one-dimensional dataset with variable number of jumps. Right: boolean functions of varying rank.

## APPENDIX B  VALIDATION SET RESULTS

Experimental results for the validation set are shown in Figure 4. Maybe somewhat surprisingly, the ranking is quite different than on the test set, with `MotherNet` outperforming `TabPFN` both in rank and normalized ROC AUC. Examined datasets with at least $0.05$ ROC AUC difference between `MotherNet` and `XGBoost`, which are shown in Table 4. Overall, `MotherNet` and `TabPFN` have similar characteristics, as might be expected from the shared synthetic training data. It's notable that both outperform `XGBoost` on the same dataasets (top rows) and are both outperformed by `XGBoost` on the same datasets. The last three rows of Table4 show a particular stark failure mode of `TabPFN` and `MotherNet`, who perform at chance level on `parity5_plus_5`, which is essentially solved by `XGBoost`, and lag severely behind `XGBoost` on `teachingAssistant` and `schizo`.

Investing these datasets, we found that there are two different failure modes present. The datasets `teachingAssistant` and `schizo` have single features that are highly informative but contain strong discontinuities with respect to the target class, see Figure C. Both could be considered data leakage via an ID column, but in essence these point to the fact that discontinuous functions with many steps, and/or memorization of ID variables are not well captured by `TabPFN` and `MotherNet`. While in these particular cases, the datasets could be considered faulty, there was information included in the data that a tree-based model was able to exploit, while `TabPFN` and `MotherNet` could not; in this case discontinuous functions with many jumps in a single continuous feature.

The `parity5_plus_5` illustrated a different failure case: this dataset relies on matching boolean patterns on a subset of the columns. While Hollmann et al. (2022) showed that irrelevant features degrade the performance of `TabPFN`, removing the irrelevant features did not improve performance on `parity5_plus_5`; the issue rather seems in the inability of `TabPFN` and `MotherNet` to memorize binary patterns. Based on these two failure cases, we generated families of synthetic functions to illustrate the shortcomings. We compare `TabPFN` and `MotherNet` to two simple baselines, `RandomForestClassifier` and `MLPClassifier` from `scikit-learn` Pedregosa et al. (2011) with default parameters without tuning, see Appendix C for details. Figure 6 shows that as the complexity of the dataset increases, either in terms of jumps in a 1d function, or in terms of complexity of a boolean function, `TabPFN` and `MotherNet` degrade in performance much more quickly than the Random Forest model. The MLP is able to easily learn the boolean datasets, but not the discontinuous 1d datasets; somewhat suprisingly, given the underperformance of `MotherNet` on this task, `MotherNet` slightly outperforms the MLP. We speculate that these failure cases can be addressed in future work by including similar synthetic datasets in the prior. It might also be necessary to adopt the architecture of `MotherNet`, for example using Fourier features Tancik et al. (2020) to account for discontinuities.

| model
dataset | MotherNet | TabPFN | XGBoost | MLP-Distill | XGB - MN |
|---|---|---|---|---|---|
| KnuggetChase3 | 0.754 | 0.770 | 0.643 | 0.651 | -0.111 |
| analcatdata_apnea2 | 0.929 | 0.877 | 0.840 | 0.939 | -0.089 |
| mc2 | 0.770 | 0.767 | 0.681 | 0.726 | -0.088 |
| conference_attendance | 0.572 | 0.576 | 0.498 | 0.538 | -0.075 |
| disclosure_z | 0.576 | 0.572 | 0.509 | 0.581 | -0.068 |
| PieChart1 | 0.863 | 0.885 | 0.808 | 0.879 | -0.055 |
| disclosure_x_noise | 0.531 | 0.533 | 0.480 | 0.540 | -0.051 |
| chscase_funds | 0.695 | 0.679 | 0.644 | 0.698 | -0.051 |
| meta | 0.810 | 0.769 | 0.864 | 0.788 | 0.054 |
| analcatdata_apnea3 | 0.890 | 0.865 | 0.947 | 0.899 | 0.057 |
| tae | 0.650 | 0.675 | 0.708 | 0.699 | 0.058 |
| triazines | 0.760 | 0.772 | 0.821 | 0.731 | 0.061 |
| prnn_fglass | 0.809 | 0.852 | 0.889 | 0.825 | 0.080 |
| pm10 | 0.496 | 0.511 | 0.591 | 0.531 | 0.094 |
| pbcseq | 0.783 | 0.839 | 0.890 | 0.832 | 0.107 |
| schizo | 0.616 | 0.636 | 0.796 | 0.627 | 0.180 |
| teachingAssistant | 0.679 | 0.692 | 0.940 | 0.709 | 0.261 |
| parity5_plus_5 | 0.453 | 0.456 | 0.992 | 0.452 | 0.539 |

Table 4: Subset of validation data where there is a difference of at least $0.05$ average ROC-AUC between `MotherNet` and `XGBoost`.

## APPENDIX C    FAILURE CASE DATASET GENERATION

Inspired by the results shown in Table 4, we created two families of synthetic datasets. The first is a binary classification task on a single feature, that is distributed uniformly between 0 and 1. For each dataset that we generate, we draw 2000 samples from the uniform distribution, and $n_{\text{steps}} - 1$ cut-off points, also between 0 and 1. At each cut-off point we flip the class label. A resulting dataset for $n_{\text{steps}} = 5$ is show in Figure 8, where we show only 100 points for illustration purposes. Note that since the split into training and test data is done using an (class-stratified) i.i.d. assumption, this dataset is trivial to learn for any tree-based or neighbors-based learner. While it is possible to learn such a dataset with a neural network, this might require tuning the architecture, and using the $MLP$ with default parameters from `scikit-learn` fails to learn this data.

The other family of synthetic datasets is inspired by the `parity5_plus_5` dataset and is a random combination of boolean conjunctions of a certain rank. The training samples in all cases are all binary sequences of length 10, where each bit is one feature, hence the feature space is $X = \{0, 1\}^{10}$. The labels for each dataset are constructed iteratively using a logical disjunction of conjunctions. In every iteration, a term is created by conjoining $r$ bits chosen at random, with each bit also randomly assigned a negation or not. Terms are continually added to the disjunction until at least one-third of the samples satisfy the formula, ensuring a relatively balanced dataset. We split the dataset randomly (but class-stratified) into training and test set. This is a relaxation of the classical parity problem; for rank 1, the label would correspond simply to one of the input features and therefore should be easily learnable for any algorithm. For rank 10, the dataset is an arbitrary boolean function, which is not learnable (in the sense that seeing only the training set in expectation provides no information on the test set).

For the experiments in Figure 6, we generated 20 datasets for each rank or step, and performed five-fold stratified cross-validation for each of them.

## APPENDIX D    HYPER PARAMETER SPACES FOR BASELINE METHODS

The hyperparameters used for the baseline models discussed in Section 4 are shown in Table 5 and were tuned with HyperOpt Bergstra et al. (2011) following the setup of Hollmann et al. (2022), using random search.

| Model | Hyperparameters |
|---|---|
| Random Forest | n_estimators: randint(20, 200), max_features: choice([None, 'sqrt', 'log2']),
max_depth: randint(1, 45), min_samples_split: choice([2, 5, 10]) |
| MLP | hidden_size: choice([16, 32, 64, 128, 256, 512]), learning_rate: loguniform($10^{-5}$, 0.01),
n_epochs: choice([10, 100, 1000]), dropout_rate: choice([0, 0.1, 0.3]),
n_layers: choice([1, 2, 3]), weight_decay: loguniform($10^{-5}$, 0.01) |
| ResNet | hidden_size: choice([16, 32, 64, 128, 256, 512]), learning_rate: loguniform($10^{-5}$, 0.01),
n_epochs: choice([10, 100, 1000]), dropout_rate: choice([0, 0.1, 0.3]),
n_layers: choice([1, 2, 3]), weight_decay: loguniform($10^{-5}$, 0.01)
hidden_multiplier: choice([1, 2, 3, 4]) |
| KNN | n_neighbors: randint(1, 16) |
| XGBoost | learning_rate: loguniform($e^{-7}$, 1), max_depth: randint(1, 10),
subsample: uniform(0.2, 1), colsample_bytree: uniform(0.2, 1),
colsample_bylevel: uniform(0.2, 1), min_child_weight: loguniform($e^{-16}$, $e^5$),
alpha: loguniform($e^{-16}$, $e^2$), lambda: loguniform($e^{-16}$, $e^2$),
gamma: loguniform($e^{-16}$, $e^2$), n_estimators: randint(100, 4000) |
| Logistic Regression | penalty: choice(['l1', 'l2', 'none']), max_iter: randint(50, 500),
fit_intercept: choice([True, False]), C: loguniform($e^{-5}$, $\log(5)$) |

Table 5: Hyperparameters for each model

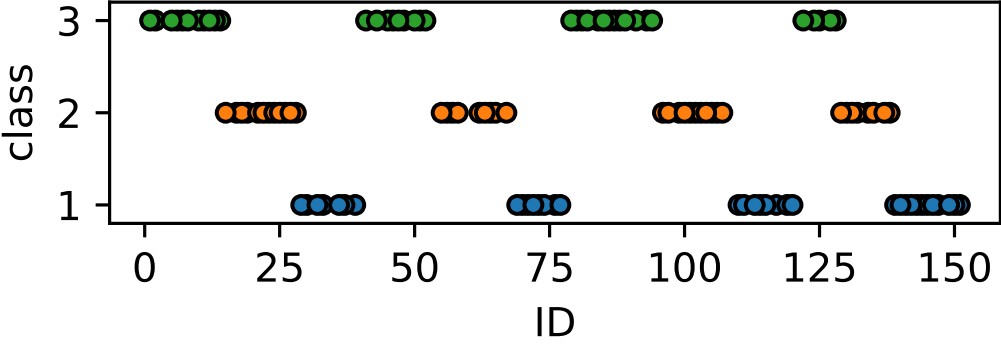

Figure 7: Class label plotted against `ID` column in `teachingAssistant` dataset shows a strong correlation that is likely data leakage.

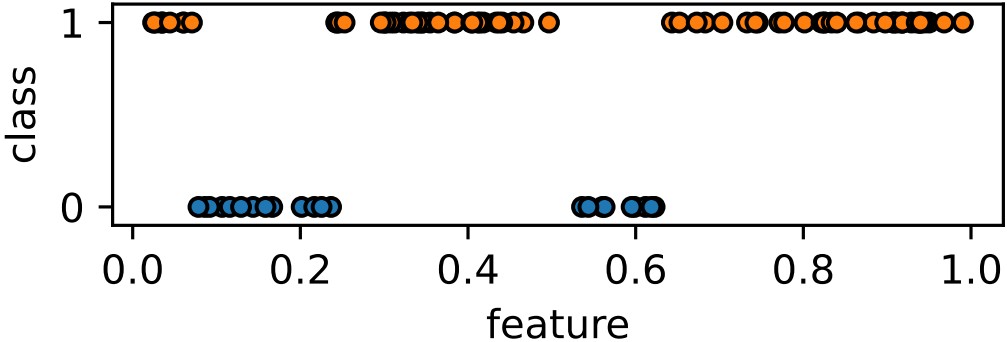

Figure 8: Example of a synthetic classification example in 1d.

| algorithm | min rank | max rank | mean rank | median rank | mean AUC |
|---|---|---|---|---|---|
| TabPFN (default) | 1 | 31 | 8.65 | 6.00 | 0.91 |
| CatBoost | 1 | 38 | 10.03 | 7.50 | 0.92 |
| CatBoost (default) | 1 | 37 | 10.89 | 9.00 | 0.92 |
| XGBoost | 1 | 32 | 11.35 | 9.00 | 0.91 |
| ResNet | 1 | 38 | 13.41 | 12.00 | 0.85 |
| SAINT | 1 | 38 | 13.50 | 11.00 | 0.86 |
| RandomForest | 1 | 37 | 13.83 | 13.50 | 0.89 |
| MotherNet (default) | 1 | 36 | 13.85 | 12.00 | 0.87 |
| XGBoost (default) | 1 | 37 | 14.10 | 13.00 | 0.89 |
| DANet | 1 | 34 | 15.22 | 14.50 | 0.85 |
| ResNet (default) | 1 | 39 | 15.92 | 16.50 | 0.82 |
| LightGBM (default) | 1 | 36 | 16.08 | 15.00 | 0.86 |
| LightGBM | 1 | 38 | 16.15 | 15.00 | 0.86 |
| RandomForest (default) | 1 | 37 | 16.55 | 14.00 | 0.85 |
| NODE | 1 | 37 | 16.99 | 17.00 | 0.83 |
| SAINT (default) | 1 | 39 | 17.05 | 16.00 | 0.82 |
| FTTransformer | 1 | 39 | 17.64 | 18.50 | 0.82 |
| SVM | 1 | 39 | 18.52 | 20.00 | 0.79 |
| MLP-rtdl | 1 | 39 | 18.54 | 17.50 | 0.77 |
| NODE (default) | 1 | 39 | 18.66 | 18.00 | 0.81 |
| FTTransformer (default) | 1 | 39 | 20.93 | 23.00 | 0.72 |
| STG | 1 | 37 | 21.27 | 23.00 | 0.73 |
| DANet (default) | 1 | 39 | 21.51 | 22.00 | 0.76 |
| MLP-rtdl (default) | 1 | 39 | 22.95 | 24.50 | 0.66 |
| LinearModel | 1 | 39 | 23.22 | 25.00 | 0.68 |
| MLP | 1 | 38 | 23.42 | 25.50 | 0.71 |
| TabNet | 1 | 39 | 24.27 | 27.00 | 0.71 |
| SVM (default) | 1 | 39 | 24.60 | 28.00 | 0.63 |
| DecisionTree | 1 | 39 | 26.89 | 28.00 | 0.62 |
| TabNet (default) | 1 | 39 | 27.13 | 29.00 | 0.63 |
| KNN | 1 | 39 | 27.82 | 29.00 | 0.62 |
| VIME | 1 | 38 | 28.31 | 31.00 | 0.60 |
| MLP (default) | 1 | 39 | 28.52 | 31.00 | 0.54 |
| DecisionTree (default) | 1 | 39 | 28.66 | 31.00 | 0.56 |
| STG (default) | 1 | 39 | 29.10 | 33.00 | 0.52 |
| KNN (default) | 1 | 39 | 29.12 | 31.00 | 0.57 |
| VIME (default) | 1 | 39 | 31.92 | 35.50 | 0.39 |

Table 6: TabZilla algorithm ranking using normalized ROC AUC, including the default configurations of all algorithms. `TabPFN` and `MotherNet` have no hyper-parameters and are therefore labeled "default".

| algorithm | min rank | max rank | mean rank | median rank | mean Accuracy |
|---|---|---|---|---|---|
| CatBoost | 1 | 19 | 5.80 | 4.00 | 0.87 |
| TabPFN | 1 | 20 | 6.14 | 5.00 | 0.83 |
| XGBoost | 1 | 18 | 7.30 | 6.00 | 0.81 |
| ResNet | 1 | 20 | 8.09 | 8.00 | 0.75 |
| SAINT | 1 | 20 | 8.35 | 7.00 | 0.73 |
| NODE | 1 | 20 | 8.38 | 8.00 | 0.74 |
| FTTransformer | 1 | 19 | 8.61 | 8.00 | 0.76 |
| RandomForest | 1 | 20 | 8.70 | 8.00 | 0.76 |
| LightGBM | 1 | 20 | 8.95 | 8.00 | 0.76 |
| MotherNet | 1 | 20 | 9.29 | 9.00 | 0.72 |
| SVM | 1 | 19 | 9.59 | 10.50 | 0.69 |
| DANet | 1 | 19 | 10.29 | 10.00 | 0.73 |
| MLP-rtdl | 1 | 20 | 10.37 | 11.00 | 0.66 |
| STG | 1 | 20 | 12.38 | 13.00 | 0.56 |
| DecisionTree | 1 | 20 | 12.43 | 14.00 | 0.59 |
| MLP | 1 | 20 | 12.65 | 14.00 | 0.57 |
| LinearModel | 1 | 20 | 12.89 | 15.00 | 0.51 |
| TabNet | 1 | 20 | 13.36 | 15.00 | 0.55 |
| KNN | 1 | 20 | 14.43 | 16.00 | 0.45 |
| VIME | 3 | 20 | 15.80 | 17.50 | 0.37 |

Table 7: TabZilla algorithm ranking using (normalized) Accuracy, compare with Table 1 in McElfresh et al. (2024)

| did | name | d | n | k | | did | name | d | n | k |
|---|---|---|---|---|---|---|---|---|---|---|
| 11 | balance-scale | 5 | 625 | 3 | | 1049 | pc4 | 38 | 1458 | 2 |
| 14 | mfeat-fourier | 77 | 2000 | 10 | | 1050 | pc3 | 38 | 1563 | 2 |
| 15 | breast-w | 10 | 699 | 2 | | 1063 | kc2 | 22 | 522 | 2 |
| 16 | mfeat-karhunen | 65 | 2000 | 10 | | 1068 | pc1 | 22 | 1109 | 2 |
| 18 | mfeat-morphological | 7 | 2000 | 10 | | 1462 | banknote-authentication | 5 | 1372 | 2 |
| 22 | mfeat-zernike | 48 | 2000 | 10 | | 1464 | blood-transfusion-... | 5 | 748 | 2 |
| 23 | cmc | 10 | 1473 | 3 | | 1480 | ilpd | 11 | 583 | 2 |
| 29 | credit-approval | 16 | 690 | 2 | | 1494 | qsar-biodeg | 42 | 1055 | 2 |
| 31 | credit-g | 21 | 1000 | 2 | | 1510 | wdbc | 31 | 569 | 2 |
| 37 | diabetes | 9 | 768 | 2 | | 6332 | cylinder-bands | 40 | 540 | 2 |
| 50 | tic-tac-toe | 10 | 958 | 2 | | 23381 | dresses-sales | 13 | 500 | 2 |
| 54 | vehicle | 19 | 846 | 4 | | 40966 | MiceProtein | 82 | 1080 | 8 |
| 188 | eucalyptus | 20 | 736 | 5 | | 40975 | car | 7 | 1728 | 4 |
| 458 | analcatdata_authorship | 71 | 841 | 4 | | 40982 | steel-plates-fault | 28 | 1941 | 7 |
| 469 | analcatdata_dmft | 5 | 797 | 6 | | 40994 | climate-model-... | 21 | 540 | 2 |

Table 8: Test dataset names and properties, taken from Hollmann et al. (2022). Here *did* is the OpenML Dataset ID, *d* the number of features, *n* the number of instances, and *k* the number of classes in each dataset.

## APPENDIX E    VALIDATION SET

We use the validation set of Hollmann et al. (2022), as listed in Table 9.

| did | name | d | n | k |
|---|---|---|---|---|
| 13 | breast-cancer | 10 | 286 | 2 |
| 25 | colic | 27 | 368 | 2 |
| 35 | dermatology | 35 | 366 | 6 |
| 40 | sonar | 61 | 208 | 2 |
| 41 | glass | 10 | 214 | 6 |
| 43 | haberman | 4 | 306 | 2 |
| 48 | tae | 6 | 151 | 3 |
| 49 | heart-c | 14 | 303 | 2 |
| 51 | heart-h | 14 | 294 | 2 |
| 53 | heart-statlog | 14 | 270 | 2 |
| 55 | hepatitis | 20 | 155 | 2 |
| 56 | vote | 17 | 435 | 2 |
| 59 | ionosphere | 35 | 351 | 2 |
| 61 | iris | 5 | 150 | 3 |
| 187 | wine | 14 | 178 | 3 |
| 329 | hayes-roth | 5 | 160 | 3 |
| 333 | monks-problems-1 | 7 | 556 | 2 |
| 334 | monks-problems-2 | 7 | 601 | 2 |
| 335 | monks-problems-3 | 7 | 554 | 2 |
| 336 | SPECT | 23 | 267 | 2 |
| 337 | SPECTF | 45 | 349 | 2 |
| 338 | grub-damage | 9 | 155 | 4 |
| 377 | synthetic_control | 61 | 600 | 6 |
| 446 | prnn_crabs | 8 | 200 | 2 |
| 450 | analcatdata_lawsuit | 5 | 264 | 2 |
| 451 | irish | 6 | 500 | 2 |
| 452 | analcatdata_broadwaymult | 8 | 285 | 7 |
| 460 | analcatdata_reviewer | 8 | 379 | 4 |
| 463 | backache | 32 | 180 | 2 |
| 464 | prnn_synth | 3 | 250 | 2 |
| 466 | schizo | 15 | 340 | 2 |
| 470 | profb | 10 | 672 | 2 |
| 475 | analcatdata_germangss | 6 | 400 | 4 |
| 481 | biomed | 9 | 209 | 2 |
| 679 | rmftsa_sleepdata | 3 | 1024 | 4 |
| 694 | diggle_table_a2 | 9 | 310 | 9 |
| 717 | rmftsa_ladata | 11 | 508 | 2 |
| 721 | pwLinear | 11 | 200 | 2 |
| 724 | analcatdata_vineyard | 4 | 468 | 2 |
| 733 | machine_cpu | 7 | 209 | 2 |
| 738 | pharynx | 11 | 195 | 2 |
| 745 | auto_price | 16 | 159 | 2 |
| 747 | servo | 5 | 167 | 2 |
| 748 | analcatdata_wildcat | 6 | 163 | 2 |
| 750 | pm10 | 8 | 500 | 2 |
| 753 | wisconsin | 33 | 194 | 2 |
| 756 | autoPrice | 16 | 159 | 2 |
| 757 | meta | 22 | 528 | 2 |
| 764 | analcatdata_apnea3 | 4 | 450 | 2 |

| did | name | d | n | k |
|---|---|---|---|---|
| 765 | analcatdata_apnea2 | 4 | 475 | 2 |
| 767 | analcatdata_apnea1 | 4 | 475 | 2 |
| 774 | disclosure_x_bias | 4 | 662 | 2 |
| 778 | bodyfat | 15 | 252 | 2 |
| 786 | cleveland | 14 | 303 | 2 |
| 788 | triazines | 61 | 186 | 2 |
| 795 | disclosure_x_tampered | 4 | 662 | 2 |
| 796 | cpu | 8 | 209 | 2 |
| 798 | cholesterol | 14 | 303 | 2 |
| 801 | chscase_funds | 3 | 185 | 2 |
| 802 | pbcseq | 19 | 1945 | 2 |
| 810 | pbc | 19 | 418 | 2 |
| 811 | rmftsa_ctoarrivals | 3 | 264 | 2 |
| 814 | chscase_vine2 | 3 | 468 | 2 |
| 820 | chatfield_4 | 13 | 235 | 2 |
| 825 | boston_corrected | 21 | 506 | 2 |
| 826 | sensory | 12 | 576 | 2 |
| 827 | disclosure_x_noise | 4 | 662 | 2 |
| 831 | autoMpg | 8 | 398 | 2 |
| 839 | kdd_el_nino-small | 9 | 782 | 2 |
| 840 | autoHorse | 26 | 205 | 2 |
| 841 | stock | 10 | 950 | 2 |
| 844 | breastTumor | 10 | 286 | 2 |
| 852 | analcatdata_gsssexsurvey | 10 | 159 | 2 |
| 853 | boston | 14 | 506 | 2 |
| 854 | fishcatch | 8 | 158 | 2 |
| 860 | vinnie | 3 | 380 | 2 |
| 880 | mu284 | 11 | 284 | 2 |
| 886 | no2 | 8 | 500 | 2 |
| 895 | chscase_geyser1 | 3 | 222 | 2 |
| 900 | chscase_census6 | 7 | 400 | 2 |
| 906 | chscase_census5 | 8 | 400 | 2 |
| 907 | chscase_census4 | 8 | 400 | 2 |
| 908 | chscase_census3 | 8 | 400 | 2 |
| 909 | chscase_census2 | 8 | 400 | 2 |
| 915 | plasma_retinol | 14 | 315 | 2 |
| 925 | visualizing_galaxy | 5 | 323 | 2 |
| 930 | colleges_usnews | 34 | 1302 | 2 |
| 931 | disclosure_z | 4 | 662 | 2 |
| 934 | socmob | 6 | 1156 | 2 |
| 939 | chscase_whale | 9 | 228 | 2 |
| 940 | water-treatment | 37 | 527 | 2 |
| 941 | lowbwt | 10 | 189 | 2 |
| 949 | arsenic-female-bladder | 5 | 559 | 2 |
| 966 | analcatdata_halloffame | 17 | 1340 | 2 |
| 968 | analcatdata_birthday | 4 | 365 | 2 |
| 984 | analcatdata_draft | 5 | 366 | 2 |
| 987 | collins | 23 | 500 | 2 |
| 996 | prnn_fglass | 10 | 214 | 2 |

Table 9: Validation dataset names and properties, taken from Hollmann et al. (2022). Here *did* is the OpenML Dataset ID, *d* the number of features, *n* the number of instances, and *k* the number of classes in each dataset.

Table 10: Validation datasets, continued

| did | name | d | n | k |
|---|---|---|---|---|
| 1048 | jEdit_4.2_4.3 | 9 | 369 | 2 |
| 1054 | mc2 | 40 | 161 | 2 |
| 1071 | mw1 | 38 | 403 | 2 |
| 1073 | jEdit_4.0_4.2 | 9 | 274 | 2 |
| 1100 | PopularKids | 11 | 478 | 3 |
| 1115 | teachingAssistant | 7 | 151 | 3 |
| 1412 | lungcancer_GSE31210 | 24 | 226 | 2 |
| 1442 | MegaWatt1 | 38 | 253 | 2 |
| 1443 | PizzaCutter1 | 38 | 661 | 2 |
| 1444 | PizzaCutter3 | 38 | 1043 | 2 |
| 1446 | CostaMadre1 | 38 | 296 | 2 |
| 1447 | CastMetal1 | 38 | 327 | 2 |
| 1448 | KnuggetChase3 | 40 | 194 | 2 |
| 1451 | PieChart1 | 38 | 705 | 2 |
| 1453 | PieChart3 | 38 | 1077 | 2 |
| 1488 | parkinsons | 23 | 195 | 2 |
| 1490 | planning-relax | 13 | 182 | 2 |
| 1495 | qualitative-bankruptcy | 7 | 250 | 2 |
| 1498 | sa-heart | 10 | 462 | 2 |
| 1499 | seeds | 8 | 210 | 3 |
| 1506 | thoracic-surgery | 17 | 470 | 2 |
| 1508 | user-knowledge | 6 | 403 | 5 |
| 1511 | wholesale-customers | 9 | 440 | 2 |
| 1512 | heart-long-beach | 14 | 200 | 5 |
| 1520 | robot-failures-lp5 | 91 | 164 | 5 |

| did | name | d | n | k |
|---|---|---|---|---|
| 1523 | vertebra-column | 7 | 310 | 3 |
| 4153 | Smartphone-Based_Re... | 68 | 180 | 6 |
| 23499 | breast-cancer-dropped-... | 10 | 277 | 2 |
| 40496 | LED-display-domain-7... | 8 | 500 | 10 |
| 40646 | GAMETES_Epistasis_2-... | 21 | 1600 | 2 |
| 40663 | calendarDOW | 33 | 399 | 5 |
| 40669 | corral | 7 | 160 | 2 |
| 40680 | mofn-3-7-10 | 11 | 1324 | 2 |
| 40682 | thyroid-new | 6 | 215 | 3 |
| 40686 | solar-flare | 13 | 315 | 5 |
| 40690 | threeOf9 | 10 | 512 | 2 |
| 40693 | xd6 | 10 | 973 | 2 |
| 40705 | tokyo1 | 45 | 959 | 2 |
| 40706 | parity5_plus_5 | 11 | 1124 | 2 |
| 40710 | cleve | 14 | 303 | 2 |
| 40711 | cleveland-nominal | 8 | 303 | 5 |
| 40981 | Australian | 15 | 690 | 2 |
| 41430 | DiabeticMellitus | 98 | 281 | 2 |
| 41538 | conference_attendance | 7 | 246 | 2 |
| 41919 | CPMP-2015-runtime-... | 23 | 527 | 4 |
| 41976 | TuningSVMs | 81 | 156 | 2 |
| 42172 | regime_alimentaire | 20 | 202 | 2 |
| 42261 | iris-example | 5 | 150 | 3 |
| 42544 | Touch2 | 11 | 265 | 8 |
| 42585 | penguins | 7 | 344 | 3 |
| 42638 | titanic | 8 | 891 | 2 |

Table 11: Validation dataset, continued

