# OpenReview forum: "MotherNet: Fast Training and Inference via Hyper-Network Transformers"
_ICLR.cc/2025/Conference — ICLR 2025 Poster_

### Official Review · Reviewer_DRbs · 2024-10-31

**Soundness:** 2
**Presentation:** 1
**Contribution:** 2
**Rating:** 3
**Confidence:** 4

**Summary:**

MotherNet performs in-context learning on tabular data via generating a feed-forward network from a pre-trained transformer to perform prediction. The work is a synthesis of ideas from TabPFN and meta-learning/hypernetworks. MotherNet’s efficiency stems from requiring only a forward pass on the pre-trained model and the simplicity of the feed-forward MLP. Experiments are conducted on the OpenML CC-18 and TabZilla benchmarks.

**Strengths:**

1. The pipeline is efficient during training/fitting and inference.
2. Empirical evaluations include an extensive set of baselines, including distillation methods, and two frequently-used benchmarks for tabular data.

**Weaknesses:**

1. MotherNet is not a competitive method based on the average rank numbers. It is outperformed on CC-18 by MLP-Distill and TabPFN. Such is the case for TabZilla, but the error intervals are too wide for any conclusions to be drawn. This experiment is not effective. Pre-training time is not rigorously compared as part of the method’s cost.
2. The paper itself is of subpar quality, especially in the presentation of results. There are too many references to appendix figures and tables, which should be discouraged and used sparingly. Paradoxically, the paper does not even take up the full page limit of 10 pages. There are errors in references (e.g. line 405 - Table 4.2). Understanding the results is very difficult. The critical difference diagrams are not explained.
3. Novelty is limited as the method is heavily influenced by TabPFN. It could be seen as an extension.

**Questions:**

I strongly encourage the authors to restructure Section 4 and rearrange the results. Also, a stronger argument is necessary to show why MotherNet is better than tree-based methods and TabPFN.

---

> ### Public Comment · ~Yunc_G1 · 2024-11-13
>
> This is a very fair and insightful review, demonstrating the reviewer’s profound understanding and expertise in this field.

---

> ### Author Response · Authors · 2024-11-21
> **Author Response**
>
> Thank you for your comments.
> We want to emphasize that we do not claim "MotherNet is better than TabPFN and tree-based methods", so this is not an argument that we would like to defend. MotherNet overcomes one of the limitations of TabPFN, while still providing competitive performance. We would like to claim that we show that on small datasets, MotherNet is highly effective without any dataset specific tuning, and that in-context learning of MLPs using Mothernet outperforms gradient descent training of similar models even if hyper-parameter tuning is performed.
>
> ### weaknesses
> 1) How do you suggest we include pre-training time into the methods cost? Generally, per-dataset training cost is considered. The baseline methods benefit from years of algorithm tuning by a large community of practitioners; that is usually also not included in the per-dataset cost, as it is not relevant to applying the model to a new task.
> We used two established benchmarks for evaluating our algorithm. Since you conclude this is not an effective experiment, what alternative would you suggest? It would be easy to create a benchmark that favors our method, however, we opt to provide a broad comparison based on established benchmarks; that the results are not statistically significant points to the hardness of rigorous testing and the lack of appropriate benchmarks – unless you can suggest a different benchmark, that you judge would be more effective. Our conclusion is that given the existing benchmarks, our method is statistically equivalent to the top performing methods, while providing extremely fast training and prediction speeds, using a novel approach that we think is noteworthy as it outperforms well-established gradient descent training.
>
> 2) If you could be more specific in your criticism, we are happy to address your concers about the presentation. We are happy to address the errors in the references. All the main results are included in the paper, and the figures in the appendix are additional results. We are happy to move some of the figure from the appendix to the main paper if you prefer; this seems a simple matter. To the best of our knowledge, the critical difference diagram is the main way the ML community compares algorithms across datasets. We are happy to provide a review of the methodology in the final version.
>
> 3) This work indeed is an extension of TabPFN, however, in a way that we think is interesting, that overcomes a limitation of TabPFN which TabPFN has been criticized for (long prediction time) and in a way that was consider impossible to achieve by the TabPFN authors (as per personal communication).

---

> ### Comment · Reviewer_DRbs · 2024-11-21
> **Response**
>
> To justify my original short review, I believe the work suffers from a fundamental weakness and a lesser drawback: performance results and novelty. It is argued that MotherNet is "competitive," but the performance numbers contradict this claim. On CC-18, the normalized AUC is worse than TabPFN, and on TabZilla if one disregards the wide error bars, MotherNet comes in seventh for average rank. Reviewer DaxP and reviewer QK8H share a similar opinion. On the other hand, the fact that the method is an incremental one detracts from the contribution of the paper. The contribution of an interesting result (reviewer QK8H) is below the qualification of ICLR as a top conference. Note that it is the author's responsibility to produce convincing results on an accepted benchmark. The issue of presentation is a separate discussion. It is already pointed out that substantial arguments in Section 4 relies on results and figures in the appendix, so it is important to add necessary results to the main text. Unfortunately, there has not been a revision at this time; therefore, I maintain my evaluation.

---

> ### Author Response · Authors · 2024-11-28
> **Updated PDF**
>
> As mentioned above, the goal of this work is not to improve the accuracy of TabPFN, but to improve the inference speed. We consider outperforming a tuned gradient boosting model a competitive approach; we do not claim setting a new state of the art of accuracy.
>
> We are happy to produce results on an accepted benchmark; the question was which benchmark you consider accepted. Our results on both CC-18 and TabZilla are quite strong, in particular given that we perform pure in-context learning, using orders of magnitude less compute than competing approaches, and no hyper-parameter tuning.
>
> We added some of the results to the main text, please let us know if that addresses your concern. We consider the validation set results supplemental but we moved them to the main text to address your feedback.
>
> It seems you are concerned that the novelty of the architecture and the fact that MotherNet is ranks below TabPFN in terms of accuracy are your main concern. What evidence or experiment would convince you that the improved prediction time is a trade-off that can be beneficial?

---

### Official Review · Reviewer_DaxP · 2024-11-03

**Soundness:** 2
**Presentation:** 3
**Contribution:** 3
**Rating:** 6
**Confidence:** 4

**Summary:**

This study proposes a classification hypernetwork for tabular data, called MotherNet. A hypernetwork is a "parent" network that, given training data, produces the weights for another "child" network that is immediately ready to make predictions on the test data. This allows completely avoiding the traditional gradient-based training and hyperparameter tuning.

To learn how to produce good child networks, MotherNet itself is trained on synthetic data for a long time (28 days on one GPU A100). The synthetic data generation is inherited from the TabPFN project. The parent network is a 12-layer Transformer, same as TabPFN. The child network is a lightweight 2-layer MLP. Since the actual predictions are made by the child network, it means that the inference efficiency is high.

On datasets of size <5000 objects, a competitive performance of MotherNet is reported.

**Strengths:**

I like the overall idea of the project. This paper shows an interesting new path with certain benefits over the original TabPFN, including the high inference efficiency.

*(Assuming that the authors will share the code and the model checkpoints linked in the paper)*
To me, a big positive thing is the provided code for training MotherNet and the model weights. Training hypernetworks is non-trivial and costly, and I like that this project gives the community a ready-to-use hypernetwork baseline.

Continuing the previous point, the proposed model seem to outperform HyperFast, one of the previous (and the only?) tabular hypernetwork. So MotherNet advances the niche of tabular hypernetworks, and gives a better starting point for future work in this direction.

I also appreciate that the paper explicitly communicates the scope of dataset sizes (<5000 objects), as well as the analysis of some failure cases in Appendix.

**Weaknesses:**

**Benchmarks**

I have mixed feelings about the benchmarks. I tend to agree that MotherNet outperforms HyperFast, however, the overall ranking of models is less convincing to me. I can imagine that the relative performance of the methods will change significantly in a different setup, and especially on different datasets.

Regarding datasets, if I understand correctly, there are three parts of results:

- (Figure 2 and Table 1) The performance on datasets with unusual ranking of models, in particular with the high performance of a linear model, as directly noted in the paper on L357. It is unclear how reliable are the conclusions made on these datasets.
- (Figure 6) The performance on the "validation" datasets. My understanding is that these results are somewhat additional, and thus presented in Appendix, not in the main text. I should admit I do not fully understand the role of the validation datasets, so I may be missing something.
- (Table 2, Figure 3, Table 6) The results on the TabZilla benchmark. There are also certain unusual results, e.g. the linear model outperforming MLP, or SVM outperforming MLP-rtdl.

Regarding hyperparameters:
- I noticed that the number of training epochs for MLP and ResNet is tuned within the grid (10, 100, 1000). I can imagine that on some datasets all three values are suboptimal, e.g. 10 can lead to underfitting, and 100 and 1000 can lead to overfitting.
- If the "choice" distritbution for HyperOpt is a categorical distribution that is order-unaware, then this can be misleading for the hyperparameter tuning engine (hard to tell to what extent).

**Related work**

Section 2 (related work) does not cover a whole family of methods that, I think, is highly relevant. However, this seems to be easy to fix, plus, some of the methods are already used as baselines. Namely, I imply the traditional machine learning models that should be trained from scratch for each task. That includes classic ML algorithms (linear models, tree-based methods, etc.) and DL architectures (there are too many of them to list; I suggest taking any popular baseline, e.g. FT-Transformer from the same paper as the already used ResNet, and traverse the citation graph backwards and forwards).

**Analysis**

In my opinion, the analysis of the MotherNet's performance should be extended. I suggest the following experiments:

(1) Finetuning the child network produced by MotherNet. Even if it is not how MotherNet is supposed to be used in practice, it would show the full potential of the MotherNet system. If this experiment is not presented in this paper, then it can become a somewhat must have experiment for future researchers willing to use MotherNet as a baseline. I recommend conducting this experiment directly in this submission.

(2) Extensive hyperparameter tuning (learning rate, weight decay, dropout, etc.) for a plain MLP of the *exactly* same architecture as the child network. Otherwise, it is unclear how optimal are the weights produced by MotherNet. For example, if the traditional tune-and-train approach gives significantly better results for the child architecture, then it can be the preferable approach when the task performance is more important than the (almost) zero cost of the MotherNet's forward pass (plus, training on small datasets is cheap).

**Questions:**

-

---

> ### Author Response · Authors · 2024-11-21
> **Author Response**
>
> We would like to thank reviewer DaxP for their comments and suggestions.
> We want to point out that the submission has an anonymized github link that contains the full code for training and prediction, as well as code to download pre-trained weights. We will address the other feedback below.
>
> ### Benchmarks
> There are indeed three results, two main experiments and additional results on the tabpfn validation set. To clarify the validation set, this methodology is common in AutoML work, where there is a chance to overfit model selection or hyper-parameters to specific datasets. All the development of MotherNet (and TabPFN), i.e. architecture, learning rates, prior, etc, have been done on the validation datasets, and only a final evaluation has been done on the test dataset. This is often not done in ML outside the AutoML community and can lead to overly optimistic benchmark results. Therefore the validation set is relegated to the appendix; however, it’s likely that most competing methods where also developed including the datasets in our test set.
>
> We chose to evaluate on two benchmarks, exactly because for any method, the relative performance will always change depending on the benchmarks. We provide two benchmarks to provide a broader picture, instead of providing a strong message on a cherry-picked selection. We used two widely available benchmark datasets; as mentioned above, if you have another benchmark dataset that you would find more convincing, we can try to include it; however, doing so is extremely expensive for all competing methods (but not for mothernet). We could also consider the AutoML benchmark or the Grinsztajn datasets, whoever both focus more on larger datasets and are therefore less relevant to this work.
>
> The results from Table 2 and  Figure 3 and Table 6 that are not for MotherNet are taken directly from the TabZilla benchmark which has been published in the NeurIPS benchmark track. We obtained the raw numbers from the authors of that work. If these numbers seem, I don't think that can be considered a shortcoming of our work.
>
> ### Hyperparameters
> We tried to adhere to the methodology from the TabPFN paper for CC-18 to be more directly comparable; "Revisiting Deep Learning Models for Tabular Data" for example uses a much broader search space, which is what was used for the TabZilla benchmark (though potentially with less budget, as described in the TabZilla paper).
> We are using randomized search as in the TabPFN paper, so ordering is indeed ignored.
>
> ### Related work
> We are happy to discuss classical algorithms like tree-based ones in the related work section; we compare against tree-based algorithms and deep learning baselines like FT-Transformer in Table 2, we are happy to also mention them in the related work section, though a thorough review of all of tabular ML is out of the scope of the paper.
>
> ### Analysis
> 1) We apologize for not properly calling out the experiments addressing 1) in the submission. The results are actually included in Table 3 in the appendix. We found that with one hour of hyper-parameter tuning, we cannot improve upon the weights produced by the hyper-network using gradient descent. We are happy to include a discussion of this experiment in the final version of the paper.
>
> 2) We did extensive hyper-parameter tuning on a space including the architecture of the child network to produce the MLP results (minus the low-rank constraint with is an optimization that did not alter accuracy in our experiments). Are you suggesting that limiting the hyper-parameter space would provide a fairer comparison? Or are you suggesting the low-rank constraint should be included in the hyper-parameter search?

---

> > ### Comment · Reviewer_DaxP · 2024-11-23
> >
> > I thank the authors for the reply. Just in case, I clarify that I am still leaning towards **accepting** this paper.
> >
> > I also have several suggestions. They are not very small in terms the technical activities required to implement them, and I really don't know how to approach this from the perspective of keeping or raising the score, because the outcome of implementing these suggestions is hard to predict. So I just share them as ideas in case they can be useful.
> >
> > (A) I think the evaluation on the benchmark from `[1]` is worthy, and in particular seeing if MotherNet performs well on that benchmark. Comments:
> > - The idea is to use the benchmark that explicitly claims to filter out non-representative datasets. Currently, in the submissions, there are things like trivial tasks (at least six tasks solvable with 1.0 accuracy) and unusual model ranking. This can reduce confidence in the results.
> > - If the size of the datasets is an issue, they can be subsampled. Multiple subsampled versions of each dataset can make results more robust.
> > - Some datasets from `[1]` have the label leakage issue, I think they should be excluded (some of such datasets are identified in `[2]`).
> >
> > (B) I think an additional finetuning experiment using the datasets from (A) would be interesting. Comments:
> > - The idea is to play safe and answer the following question: is it possible to just slightly improve the produced child network?
> > - With that in mind, I think of hyperparameter ranges like `n_epochs: int([1, 2, 3]`) and `lr: loguniform([1e-5, 1e-4])`. I also think of setting `dropout_rate=0.0` and `weight_decay=0.0`, but I am less sure about that. In fact, if `n_epochs` and `lr` are the only hyperparameters, perhaps, a better approach is to do an exhaustive grid search and summarize results in some informative way.
> > - Continuing the above idea, perhaps it can be useful to start the finetuning with one training step over the whole dataset with zero learning rate. The idea is to initialize the momentum of the optimizer before starting the actual training.
> >
> > (C) As mentioned earlier, I think that Classic ML & DL methods should be discussed in Related work. The exhaustive overview is not expected, citing a couple of popular works is enough. The motivation is to make readers aware of such mainstream models, and position this submission w.r.t. to them.
> >
> > The below text is related to the prior discussion.
> >
> > **Benchmarks**
> >
> > In my opinion, the fact that datasets were used in prior literature works only to some extent. This is because public benchmarks are known to have various quality issues, and additional job on the side of authors may be needed to filter datasets. `[1]` is an example of this approach.
> >
> > **Analysis**
> >
> > (1) This is great that the finetuning experiment is presented, thanks for pointing to it. My intuition is that readers will be looking for this experiment, and it is worth discussing in the main text regardless of the results.
> >
> > (2) Yes, I mean limiting the space by fixing the same architecture-related hyperparameters as in the child network and tuning only the remaining hyperparameters.
> >
> > **Experiment setup**
> >
> > Generally, I hypothesize that experiments on small datasets may be very "noisy", in the sense that the selection of technical details (e.g. batch size, the number of epochs, etc.) may have significant impact on the results and conclusions. For example, this is why I pointed to the number of epochs tuned over such sparse grid. In particular, such details may be important for the both analysis-related experiments mentioned above.
> >
> > **References**
> >
> > - `[1]` Why do tree-based models still outperform deep learning on tabular data?
> > - `[2]` TabReD: Analyzing Pitfalls and Filling the Gaps in Tabular Deep Learning Benchmarks

---

> > > ### Author Response · Authors · 2024-11-26
> > >
> > > Thank you for the suggestions. Regarding (A), this would certainly be interesting. The main problem with additional experiments is not to run MotherNet, which is near instantaneous, but to provide "fair" baselines which take tremendous computational resources for training and parameter tuning. Therefore we opt to reuse existing benchmarks as much as possible. Given more time, it might be worth spending the many GPU-days required to tune MLPs on a benchmark such as a subsampled version of [1].
> > >
> > > (B) This is indeed an interesting experiment; We have done a formal full search over a much larger space and include the results in Table 3 in the appendix. We have spend some time manually trying to "carefully" tune with low number of epochs and/or learning rate, but we could not find an improvement on the datasets we experimented with. Some cases, like the failure cases described in the appendix, are likely to allow for improvements, though here instead of careful tuning, essentially we are learning a new model from scratch. The space we used for hyper-parameter search did not focus on small epochs but did include small epoch numbers and small learning rates - note that each epoch is likely to be a single gradient update given the dataset size.
> > > Overall, we do not expect gradient descent to help in the small dataset setting, since it overwrites any regularization learned by the transformer, which is why we did not experiment with fine-tuning in too much detail.  Fine tuning might be more beneficial for larger or more difficult datasets.
> > >
> > > (C) Happy to include the discussion of classical algorithms in the final version.
> > >
> > > Benchmarks:
> > > We agree that the benchmark landscape is evolving; [1] in particular was selected as datasets on which tree-based models have an edge over logistic regression, which is why it is often considered the "tree friendly" benchmark. Including it would definitely broaden the analysis.
> > >
> > > Analysis
> > > It was definitely an oversight not to include the fine-tuning experiment in the main paper, thank you for pointing it out.
> > >
> > > Experimental Setup
> > > It is plausible that the results are noisy for smaller models; however, that draws attention to a big benefit of MotherNet, which is stability. We used 1h for hyper-parameter tuning per dataset, and a finer grid on such small datasets is likely to lead to a more noisy evaluation (by focusing on noise on the small validation set). With much less computational resources, MotherNet is able to provide robust results without any hyper-parameter tuning.
> > >
> > > Thank you again for all your valuable suggestions for improving the paper.

---

### Official Review · Reviewer_QK8H · 2024-11-03

**Soundness:** 2
**Presentation:** 4
**Contribution:** 3
**Rating:** 6
**Confidence:** 4

**Summary:**

This paper shows that in-context tabular models can be trained to generate MLP weights directly. The resulting MLPs generally outperform standard trained MLPs and are competitive with other tabular predictive models. The paper focuses on small datasets due to limitations of the in-context modelling being used.

**Strengths:**

- The fact that this approach works well is surprising and compelling. It appears that training on synthetic datasets has a useful regularizing effect in generating MLPs compared to standard MLPs that are just trained on the dataset in question.

- The presentation is generally very clear.

**Weaknesses:**

- The experimental results aren't very compelling on the whole. The CC-18 evaluation is limited to very small datasets and the actual differences in AUC between the top ten models are very small. Given the scale of the data and relative effectiveness of very simple models like logistic regression, it's tough to see the computational efficiency of the proposed model as actually mattering in practice, especially since it's using GPU hardware. The Tabzilla evaluation provides a wider range of dataset sizes, but in this regime, the proposed method performs about as well as an untuned XGBoost, and significantly worse than tuned GBDT models (in addition to TabPFN).

- I think the use of ensembling also weakens the experimental results. While TabPFN did this in their paper, it's generally the case that ensembling any stochastically trained model can improve performance, and I think that using ensembling on some methods and not others isn't a fair comparison.

- I'm not confident about the contributions of this paper beyond an interesting result - I'd like to hear more from the authors on that (see below).

**Questions:**

- Since the model could handle 30,000 data points on GPU and 100,000 data points on CPU, why didn't you evaluate on larger datasets (or in the case of Tabzilla, with a larger training set)?

- In general, how do you see this paper contributing to future research or practical applications? I think it's interesting that this sort of model is possible, but I'm having trouble seeing what the broader contributions are, especially because I'm not convinced the experimental results show a practical niche for the model as proposed.

- What are the main contributions of this paper compared to Hyperfast specifically? The performance of the two models is only compared on very small datasets. But beyond that, I'd like to know how you'd compare this work to Hyperfast because a comparison isn't given in Section 2.

Minor issues:
- On page 5, how did you get the number 25,738? I calculate the number of low-rank weights as 33,088 ($2hr+Nr$).
- In Section 3.1, $r$ is used to represent a rank whereas in Section 3.2, it's used for the number of input features, which is a bit confusing.
- On page 6, the text says that MotherNet outperforms MLP-Distill in terms of normalized AUC, but the results table only shows it outperforming in terms of raw AUC.
- There are a few references to "Table 4.1" and "Table 4.2" that are incorrect.

---

> ### Author Response · Authors · 2024-11-21
> **Author Response**
>
> We would like to thank reviewer QK8H for their comments and suggestions, which we'll address below.
>
> ### Why limiting to small datasets
> While we were able to computationally handle larger datasets in inference, the training data included up to 2000 points. While we expect the model to generalize somewhat beyond this (as shown in the TabPFN paper), we think it’s unlikely to scale to orders of magnitude more training data and perform well. Training on much larger datasets would be quite expensive with the current procedure.
>
> ### Practical applicability and future research
> As you say, that the approach works is somewhat surprising and compelling. This is a first step towards the full potential of the approach, and we think further research can improve accuracy much beyond what we demonstrate, for example by using other priors, other encodings and other architectures. As with all foundational models, training the model is quite expensive, which makes it difficult to fully study the effects of modifications in a single paper. This paper is meant as a feasibility study for the approach. We also disagree with the absence of a niche; there are many datasets we studied (unfortunately many of them in the validation set) on which linear models do not perform well, and MotherNet excels. Given the very good performance of TabPFN, it is likely that this model can be improved; on small datasets it already out-performs the default configurations of GBRT implementations that have a decade of tuning behind them.
>
> ### Hyperfast
>
> The main contribution beyond hyperfast is that this model is accurate on small datasets, and does not require hyper-parameter tuning and gradient descent. While they hyperfast paper discusses gradient descent and hyperparameter tuning to be optional, in practice they are not, as shown by our results. Hyperfast without hyper-parameter tuning will fail catastrophically on iris, depending on the specific split chosen (this has been observed by others, but has not been published so far – we are happy to add a more detailed analysis of the many ways in which HyperFast fails in the final version). As we discuss in the paper, HyperFast is outperformed on CC-18 by essentially all baseline algorithms, even though most of the CC-18 datasets **are in the HyperFast training set**. As an added benefit, the models generated by MotherNet are orders of magnitude smaller than those generated by HyperFast. In summary, MotherNet shows that accurate hyper-networks for tabular data are possible without per-dataset gradient descent, which HyperFast alluded to but does not actually provide. MotherNet outperforms HyperFast consistently at 25,000x less per-dataset computational cost compared to HyperFast, as mentioned in the paper. We could provide an in-depth comparison to the architecture of HyperFast, but since we have not been able to produce reasonable results with HyperFast (as observed by other researchers), even on its training set, the specific architectural choices made by HyperFast seem less interesting.
>
> ### Embedding vector size
> The embedding vector size in the paper is indeed wrong, thank you for pointing that out. The correct number should be 2hr + Nh=37888  (since the final layer is not low-rank, since 10<32 having low-rank would not make much sense).
>
> ### Experimental results
> As you observe, the (also untuned) model outperforms the untuned XGBoost model, while operating under a severe limitation of only accessing at most 3000 datapoints - and training and predicting faster than XGBoost in most cases. This is unmatched by any other deep learning architecture as far as we are aware (TabPFN predictions are much slower than XGBoost).

---

> > ### Comment · Reviewer_QK8H · 2024-11-25
> >
> > Thank you for the response, just a few follow-ups:
> >
> > I understand that training with larger datasets might not have been feasible, but why not evaluate inference with larger samples than 3000 data points when evaluating on Tabzilla, since that is possible? I would expect it to be slower but still fast enough to run, and also to be the most reasonable way to apply this model to larger datasets. You describe subsampling data as a "severe disadvantage", but it's one that doesn't seem necessary for the majority of Tabzilla datasets.
> >
> > Thank you for clarifying the comparison between this work and Hyperfast. As I understand it, what you found was that Hyperfast actually does not have solid one-shot performance whereas your method does, which I agree would be a significant improvement. If so, it would be useful to note this in the Related Work or Introduction section to indicate the contribution of this work relative to Hyperfast.
> >
> > To clarify my comments about untuned XGBoost models: XGBoost hyperparameters are typically tuned per-dataset and it's not generally expected for the default parameters to perform well, so I don't think having similar performance to default XGBoost is a positive finding. (Unlike XGBoost, CatBoost does promote itself as having default parameters that tend to work well, which matches your findings, with the default CatBoost tending to outperform MotherNet.)
> >
> > On the whole, I appreciate the significance of the improvement over Hyperfast and the possible significance of this work as a basis for further research, and will take those into consideration, but my concerns about the experiments and results still remain.

---

> > > ### Author Response · Authors · 2024-11-26
> > >
> > > It would be possible to run on more data points for TabZilla datasets; since training isn't possible on datasets of this size, we expect the additional data not to help that much, but it is an interesting experiment that are happy to include in the final paper - however, the rebuttal phase does not provide enough time to perform it. We followed the approach taken in the TabZilla paper for TabPFN to have a more direct comparison to published results.
> > >
> > > > If so, it would be useful to note this in the Related Work or Introduction section to indicate the contribution of this work relative to Hyperfast.
> > >
> > > Thank you, that is a good suggestion, and we will highlight that contribution more clearly.
> > >
> > > > I don't think having similar performance to default XGBoost is a positive finding
> > >
> > > We would respectfully disagree with this assessment. MotherNet is faster than XGBoost in training and prediction and provides better performance with default parameters. This is not true for any other recently proposed deep learning method for tabular data to the best of our knowledge, therefore advancing the state of deep learning methods for tabular prediction.

---

> > > > ### Comment · Reviewer_QK8H · 2024-11-30
> > > >
> > > > Thank you for the discussion and for the paper updates.
> > > >
> > > > I'm increasing my score to a 6 because the discussion has led me to be more positive about the contribution of the paper. The experiments in the paper indicate that in some tabular data settings, MotherNet produces networks that generally have solid performance with no gradient-based training. Since the experiments also show that HyperFast produces networks that require fine-tuning for similar performance, this is a novel capability to the best of my knowledge, and significant enough in establishing this direction of research to recommend acceptance.
> > > >
> > > > My reasons for not assigning a higher score are:
> > > > - I am still doubtful about the practical utility of the model as-is. The authors have responded to this and while I respect their position, I think we'll remain in disagreement on this point.
> > > > - I still find the experiments to be limited in ways that make them less than compelling, for the reasons given in my initial review and the lack of any MotherNet results beyond 2000 points.
> > > > - I still find the use of ensembling in experiments inappropriate (a greater degree of ensembling is used for MotherNet than other non-GBDT models), and did not receive a response on this point.

---

> > > > > ### Author Response · Authors · 2024-12-02
> > > > >
> > > > > Thank you for thoughtful in-depth comments and for increasing your score based on our responses.
> > > > > Please let us address the last point about ensembling, as we failed to address it before. For the modelling approach of TabPFN that we adopt, ensembling is extremely effective since the procedure is not invariant with respect to the ordering of feature or classes; therefore introducing variability along these axes greatly decreases variance. Applying the same kind of ensemble to traditional ML models that are trained from scratch would be equivalent to ensembling multiple models with different random seeds, as they are invariant to these transformations by design. This might yield better results in some cases, but at much increased computational cost. The quantile encoding and one-hot-encoding could potentially have beneficial effects for some of the models, and it would be interesting to investigate if other models similarly benefit from these ensembles.
> > > > > We want to point out that in the revised version we submitted last week, Figure 2, right, shows results of MotherNet without ensembling, which show MotherNet still ourperforms the MLP in terms of median rank, but not outperforming XGBoost. Unfortunately did not report results without ensembling on the TabZilla benchmark.
> > > > >
> > > > > Regarding practicality, we want to make sure that we understand your point correctly. Is it that you doubt that there is a benefit in performing much faster training and prediction on small dataset? Comparing untuned CatBoost and MotherNet on TabZilla, while CatBoost has higher median AUC and rank, it is much slower (though the main comparison we give is CatBoost on CPU vs MotherNet on GPU).
> > > > > Figure 5 (right) shows that MotherNet is statistically equivalent with the top performing models (with those models being tuned) while being much faster and not requiring tuning, though we appreciate that the wide bands of statistical equivalence might make this result less compelling. Would you consider the results practically relevant only if the untuned performance of MotherNet matched that of CatBoost while providing significant speed-ups, and/or if the same speed-ups could be achieved at large dataset sizes?
> > > > >
> > > > > Thank you again for your thoughtful comments.

---

> > > > > > ### Comment · Reviewer_QK8H · 2024-12-03
> > > > > >
> > > > > > To summarize my doubts about practicality, I was referring to my points and the discussion above about settings where linear models or GBDTs with no HPO were competitive with the proposed model, plus the limitation to very small training sets. Because of the very small training sets, I don't think fit time is that significant a metric in practice (especially since the GPU usage generally rules out hardware-limited environments). On the whole, I think the results only suggest a narrow range of possible practical settings where a data scientist might have a clear preference for this method over existing established methods. I find the significance of the paper to be stronger in that it demonstrates novel capabilities that are in the ballpark of top methods in these tabular settings, and is thus a significant step towards future work in this area, rather than being a method that's particularly compelling for practical use as-is.

---

> > > > > > > ### Author Response · Authors · 2024-12-03
> > > > > > >
> > > > > > > Thank you for the clarification and summary. We appreciate your feedback.

---

### Official Review · Reviewer_B59Z · 2024-11-03

**Soundness:** 4
**Presentation:** 4
**Contribution:** 4
**Rating:** 8
**Confidence:** 4

**Summary:**

The authors propose a novel transformer based hypernetwork model that can be 'in-context' prompted with a supervised dataset and it can generate weights for a small neural network that can generalize well on this new task. Their approach borrows ideas from meta-learning, hypernetworks, transformers and distillation, and demonstrates a modern and effective way to achieve large speed ups when a task-specific task model is trained, while also retaining the generalization capability of a full training run.

Overall I find the creativity, elegance and rigor demonstrated in this paper very refreshing and commendable.

I have a bunch of questions, and no paper is without weakness, but I find the work a reasonable and worthwhile contribution to the field.

**Strengths:**

**Strengths**:
- **Creative Approach**: MotherNet is an inventive application of hypernetworks and transformer-based architectures, demonstrating how in-context learning can effectively generate task-specific models without gradient descent.
- **Efficiency and Speed**: The method achieves significant inference speed improvements over TabPFN, offering practical advantages for use cases requiring fast, on-demand predictions.
- **Eliminates Hyper-Parameter Tuning**: MotherNet operates without per-dataset hyper-parameter tuning or gradient-based training, simplifying the pipeline for real-world applications.
- **Strong Empirical Evidence**: The paper provides a thorough comparison with a range of baselines, including TabPFN, HyperFast, and traditional models like XGBoost and Random Forests, illustrating MotherNet’s consistent performance across benchmarks.
- **Open-Source Contribution**: The work includes an open-source implementation, supporting reproducibility and facilitating future research in the field.
- **Thoughtful Methodology**: The architecture decisions, such as low-rank decomposition of weights, are well-documented and provide insights into balancing memory efficiency and model capability.
- **Solid Trade-off Analysis**: The authors give a clear breakdown of the trade-offs in terms of training and inference time, presenting scenarios where MotherNet excels over other approaches.

**Weaknesses:**

- **Scalability Constraints**: MotherNet, like TabPFN, is bound by the quadratic memory requirements of transformers, limiting its usability for datasets larger than around 5,000 samples. This could be viewed as a major limitation for more extensive applications.
- **Presentation Gaps**: Some sections, particularly in the methodology and results, could benefit from improved clarity and structure to enhance readability and understanding.
- **Comparison Depth**: Although the evaluation includes multiple models, comparisons with newer, more diverse architectures would strengthen the paper’s impact and situate MotherNet more firmly in the broader ML landscape.


And my favourite 'weakness'
- **Limited Domain Exploration**: The paper focuses solely on tabular data, leaving questions about whether MotherNet’s advantages could extend to other data modalities like text or images. This work could be a neat contender for in-context learning paradigms across the board. Hypernetworks are an intriguing idea.

**Questions:**

1. **Scalability Constraints**: While MotherNet's performance on small datasets is impressive, it shares the quadratic memory limitations of standard transformers, restricting its use to smaller datasets. Have you considered modifying MotherNet to incorporate larger context or memory-efficient transformer architectures, such as those designed for long-context handling or reduced attention complexity? How might these adaptations impact its scalability and performance?

2. **Limited Domain Exploration**: While the focus on tabular data showcases MotherNet’s strengths, its hypernetwork and in-context learning capabilities seem promising for other types of data, such as text or images. Do you see potential for adapting MotherNet to these modalities? If so, what adjustments would be necessary to leverage its in-context learning for different types of tasks?

---

> ### Author Response · Authors · 2024-11-21
> **Author response**
>
> Thank you for your feedback and questions.
> It would be great if you could provide more details about where you find gaps in the presentation, so that we could close them.
>
> Comments regarding your specific questions and criticism below.
>
> ### Limited Domain Exploration
>
> Exploring other domains is certainly interesting, though out of the scope of this work. Pretraining has been extremely successful for textual and image data, so from-scratch learning of new models is rarely if ever done, in contrast to tabular data. A possible angle would be to generate task-specific heads on pre-trained models; however, that might already be possible with the mothernet architecture we provide.  Tabular data has the benefit that comparatively small models can be made to perform well, and the issue is mostly around regularization. For other modalities, models are usually much larger, and generating them from scratch would result in even larger hypernetworks.
>
> ### Scalability Constraints:
>
> We are investigating alternative attention approaches with promising results. However, the results are not at the stage yet where they could be included in the paper.
>
> ### Comparison Depth:
>
> We provide a comparison with results on TabZilla, which includes a wide variety of recent deep learning models. Is there specific models that you’d like to see added to the comparison? We do not compare against these models on the CC-18 dataset since the hyper-parameter tuning would be prohibitively expensive, and restrict our experiments only to the most similar hyperfast model.

---

### Author Response · Authors · 2024-11-21
**Author Response**

We want to thank all the reviewers for their valuable feedback, which will help us improve the submission.
We want to call out that many of the reviewers find the work interesting:
- Overall I find the creativity, elegance and rigor demonstrated in this paper very refreshing and commendable. (B59Z)
- The fact that this approach works well is surprising and compelling. (QK8H)
- This paper shows an interesting new path with certain benefits over the original TabPFN, including the high inference efficiency. (DaxP)

We hope this method can be an interesting contribution, addressing one of the weaknesses of TabPFN, demonstrating how in-context learning can outperform gradient descent training and outperforming the similar HyperNet architecture without necessarily toping all benchmark lists. The goal of this work is not to outperform TabPFN, but to illustrate a possible path to overcoming one of its weaknesses, and proof the feasibility for pure in-context learning for tabular data.

We agree that the empirical results do not show a very clear ranking of models (apart from TabPFN leading). However, we disagree that this is a weakness of our work. We used two established benchmarks from the community, and performed a rigorous meta-validation and meta-test set split that is common in the AutoML community but unfortunately not broadly adopted in the wider ML community. In fact, it is hard to find two published benchmarks that agree on algorithm rankings, and by including two, we are aiming at providing a broader picture. We are happy to consider further benchmarks suits if the reviewers have other suggestions; however, two is double the amount of most papers that we are familiar with. Two reviewers asked for a broader comparison with other deep learning methods; we want to point out that the TabZilla benchmark contains a wide variety of recent deep learning methods, and a comparison is shown in table 2.

---

> ### Author Response · Authors · 2024-11-28
> **Updated PDF**
>
> We updated the PDF to move more of the figures into the main body of the paper, highlight the main distinction over hyper-fast, and elaborate on the fine-tuning experiments for MotherNet in the main text.

---

### Meta-Review · Area_Chair_Y4EN · 2024-12-20

**Metareview:**

The paper introduces a hypernetwork for in-context learning on tabular datasets, inspired by TabPFN and meta-learning paradigms. MotherNet generates task-specific models via a pre-trained transformer, allowing for inference without gradient descent or hyperparameter tuning. The experiments focus on small datasets (<5,000 samples) and demonstrate competitive performance against strong baselines, such as TabPFN and HyperFast, while achieving significant inference speed gains. The accuracy of MotherNet is lower than that of TabPFN and tuned GBDT models. The benchmarks, while rigorous, reveal narrow settings where the model is practical at the moment. There are also some concerns about technical novelty compared to TabPFN. Overall 3 of the reviewers were positive about the method and in particular considered that the direction is promising with a sufficiently strong initial demonstration for a challenging area.

**Additional Comments On Reviewer Discussion:**

Reviewer DRbs maintained skepticism about MotherNet’s incremental contribution and its lackluster performance relative to TabPFN. Reviewer QK8H, initially hesitant, acknowledged the novelty of MotherNet’s one-shot learning and increased their score post-rebuttal. Reviewer DaxP supported acceptance, noting the engineering effort and foundational potential for future research. The authors addressed major concerns through additional experiments and clarifications

---

### Decision · Program_Chairs · 2025-01-22

Accept (Poster)